# Comparative characterization of flavivirus production in two cell lines: Human hepatoma-derived Huh7.5.1-8 and African green monkey kidney-derived Vero

**Kyoko Saito**[1]*, **Masayoshi Fukasawa**[1], **Yoshitaka Shirasago**[1], **Ryosuke Suzuki**[2], **Naoki Osada**[3,4], **Toshiyuki Yamaji**[1], **Takaji Wakita**[5], **Eiji Konishi**[6], **Kentaro Hanada**[1]

1 Department of Biochemistry and Cell Biology, National Institute of Infectious Diseases, Shinjuku-ku, Tokyo, Japan, 2 Department of Virology II, National Institute of Infectious Diseases, Musashi-murayama-shi, Tokyo, Japan, 3 Faculty of Information Science and Technology, Hokkaido University, Sapporo, Hokkaido, Japan, 4 Global Station for Big Data and Cybersecurity, GI-CoRE, Hokkaido University, Sapporo, Hokkaido, Japan, 5 Department of Virology II, National Institute of Infectious Diseases, Shinjuku-ku, Tokyo, Japan, 6 Research Institute for Microbial Diseases, Osaka University, Suita, Osaka, Japan

* saitok@nih.go.jp

## Abstract

The *Flaviviridae* is a family of enveloped viruses with a positive-sense single-stranded RNA genome. It contains many viruses that threaten human health, such as Japanese encephalitis virus (JEV) and yellow fever virus (YFV) of the genus *Flavivirus* as well as hepatitis C virus of the genus *Hepacivirus*. Cell culture systems highly permissive for the *Flaviviridae* viruses are very useful for their isolation, propagation, and diagnosis, an understanding of their biology, and the development of vaccines and antiviral agents. Previously, we isolated a human hepatoma HuH-7-derived cell clone, Huh7.5.1–8, which is highly permissive to hepatitis C virus infection. Here, we have characterized flavivirus infection in the Huh7.5.1–8 cell line by comparing with that in the African green monkey kidney-derived Vero cell line, which is permissive for a wide spectrum of viruses. Upon infection with JEV, Huh7.5.1–8 cells produced a higher amount of virus particles early in infection and were more susceptible to virus-induced cell death than Vero cells. Similar outcomes were obtained when the cells were infected with another flavivirus, YFV (17D-204 strain). Quantification of cellular and extracellular viral RNA revealed that high JEV production in Huh7.5.1–8 cells can be attributed to rapid viral replication kinetics and efficient virus release early in infection. In a plaque assay, Huh7.5.1–8 cells developed JEV plaques more rapidly than Vero cells. Although this was not the case with YFV plaques, Huh7.5.1–8 cells developed higher numbers of YFV plaques than Vero cells. Sequence analysis of cDNA encoding an antiviral RNA helicase, RIG-I, showed that Huh7.5.1–8 cells expressed not only a full-length *RIG-I* mRNA with a known dominant-negative missense mutation but also variants without the mutation. However, the latter mRNAs lacked exon 5/6−12, indicating functional loss of RIG-I in the cells. These characteristics of the Huh7.5.1–8 cell line are helpful for flavivirus detection, titration, and propagation.

**Data Availability Statement:** All relevant data are within the manuscript and its Supporting Information files.

**Funding:** This research was supported by JSPS (Japan Society for the Promotion of Science; https://www.jsps.go.jp/index.html) of KAKENHI Grant Numbers JP16K08362, JP18H02856, JP17H04003, and JP17K09447 to K.S., M.F., K.H., and T.W., respectively, AMED-CREST (Japan Agency for Medical Research and Development, Core Research for Evolutional Science and Technology; https://www.amed.go.jp/index.html) Grant Number JP19gm0910005 to K.H., and AMED Grant Numbers JP19fk0210020j1003 and JP19fk0210053j1001 to T.W. The funders had no role in study design, data collection and analysis, decision to publish, or preparation of the manuscript.

**Competing interests:** The authors have declared that no competing interests exist.

# Introduction

The *Flaviviridae* is a family of enveloped viruses with a positive-sense single-stranded RNA genome and contains many pathogenic viruses that threaten human health. The family includes many arthropod- and blood-borne viruses pathogenic to humans: The former is exemplified by members of the genus *Flavivirus* such as Japanese encephalitis virus (JEV), yellow fever virus (YFV), dengue virus, West Nile virus and Zika virus, whereas the latter, by a member of the genus *Hepacivirus*, hepatitis C virus (HCV). As for Japanese encephalitis, it is estimated that approximately 70,000 cases occur annually in the 24 endemic countries [1]. The burden of yellow fever in Africa is estimated to be 84,000–170,000 severe cases and 29,000–60,000 deaths per year [2]. Although approved human vaccines already exist for JEV, YFV, and dengue virus, there are no specific antiviral treatments against flavivirus infection so far.

Cell culture systems highly permissive for the *Flaviviridae* viruses have greatly contributed to their isolation, propagation, and diagnosis, an understanding of their biology, and the development of vaccines and antiviral agents. One representative cell culture system commonly used for flavivirus is the Vero cell line, which was established from African green monkey kidney cells [3, 4]. This cell line is susceptible to various types of viruses [5] as a result of the loss of the type I interferon (IFN) gene cluster [6, 7]. The Vero cell line is used as read-out cells for the plaque reduction neutralization test for dengue virus, as recommended by WHO [8], and other flaviviruses [9]. The cell line is also used as a substrate for the production of various vaccines [10], as exemplified by an inactivated vaccine against JEV (IXARO) [11].

In the case of HCV, two sublines of the human hepatoma HuH-7 cell line [12], Huh7.5 [13] and its derivative Huh7.5.1 [14], are highly permissive. These cell lines along with the HCV JFH1 isolate [15] can be used to reproduce the complete life cycle of the virus, and thereby have been widely used in HCV studies. The high permissiveness of the Huh7.5 and Huh7.5.1 cell lines is partly caused by mutational inactivation of retinoic-acid-inducible protein I (RIG-I), a cellular DExD/H box RNA helicase that senses viral RNA to mediate antiviral signaling, by an amino acid change at codon 55 from threonine to isoleucine (T55I) [16]. This mutation abolishes the interaction between RIG-I and the mitochondria-bound adapter mitochondrial antiviral-signaling protein [17], which is necessary to initiate a signaling cascade leading to the induction of type-I IFNs [18, 19]. Previously, we isolated a cell clone from Huh7.5.1 cells with greater permissiveness to HCV infection [20], because the Huh7.5.1 cells that we used had exhibited heterogeneous permissiveness. The clonal cell line designated as Huh7.5.1–8 produces 10-fold higher amounts of HCV than its parental Huh7.5.1 and shows phenotypically stable permissiveness to HCV infection, which is not changed even after 100 passages. However, the permissiveness of Huh7.5.1–8 cells for flaviviruses, which are distantly-related to HCV, remains to be investigated.

In this study, we have characterized JEV/YFV infection in the Huh7.5.1–8 cell line, compared with that in the Vero cell line as well as other HuH-7 sublines. We found that Huh7.5.1–8 cells exhibited higher virus productivity early in infection and greater susceptibility to virus-induced cell death than Vero cells. We also found that virus plaques developed in Huh7.5.1–8 cells grew rapidly or in large numbers, depending on the virus type. Additionally, the Huh7.5.1–8 cell line was shown to be a RIG-I null mutant. Based on our results, we discussed the potential usefulness of the Huh7.5.1–8 cell line in flavivirus studies.

# Materials and methods

## Cells

The Huh7.5.1–8 cell line was previously isolated from the Huh7.5.1 cell line, which had been kindly provided by Dr. Francis V. Chisari (The Scripps Research Institute, La Jolla, CA, USA).

The HuH-7 (No. JCRB0403) and Vero (No. JCRB9013) cell lines were purchased from the Japanese Collection of Research Bioresources (JCRB) Cell Bank, National Institute of Biomedical Innovation (Osaka, Japan). Huh7.5.1–8, Huh7.5.1, and HuH-7 cells were maintained at 37˚C in an atmosphere of 5% $CO_2$ in Dulbecco's modified Eagle's medium (DMEM; Wako Pure Chemical Industries, Osaka, Japan; No. 044–29765) supplemented with 10% (v/v) fetal bovine serum (FBS; Sigma-Aldrich, St. Louis, MO, USA; No. 172012-500ML), 0.1 mM nonessential amino acids (Nacalai Tesque, Kyoto, Japan; No. 06344–56), 100 U/ml penicillin G, and 100 μg/ml streptomycin sulfate (Nacalai Tesque; No. 26253–84). Vero cells were maintained at 37˚C in an atmosphere of 5% $CO_2$ in Eagle's minimal essential medium (EMEM; Wako Pure Chemical Industries; No. 051–07615) supplemented with 5% (v/v) heat-inactivated FBS, 100 U/ml penicillin G, and 100 μg/ml streptomycin sulfate. For comparative analysis, Huh7.5.1–8 and Vero cells were grown in the same DMEM-based medium as above but with heat-inactivated FBS. Before starting the experiments, Vero cells were adapted to this medium through at least eight passages. JEV production in Vero cells was comparable between the DMEM-based and EMEM-based media (S1A Fig).

## Viruses

The JEV Nakayama strain (isolated in Japan in 1935) and JEV rAT strain [21] were propagated in Vero cells. The YFV 17D-204 strain, a current vaccine strain derived from the 17D strain [22], was kindly provided by Dr. Tomohiko Takasaki (Department of Virology I, National Institute of Infectious Diseases) and propagated in Vero cells. The resultant culture supernatants were stored at −80˚C and used as virus stocks. These virus strains were chosen because they could be used in our biosafety level 2 laboratory.

## Infection and post infection conditions

Cells were seeded in 24-well plates (Corning, Corning, NY, USA; No. NCO3524) one day before infection. The cells were incubated with a virus at a multiplicity of infection (MOI) of 0.1 in 0.2 ml of culture medium with reduced serum (2%) at 37˚C for 2 h. After the virus-containing medium was removed, the cells were grown in 0.5 ml of complete culture medium. The time of virus removal was taken as 0 h post-infection (h pi) for all experiments. If a post-infection culture was extended beyond 2 days post-infection (d pi), 0.5 ml of fresh medium was added at 2 d pi. Time points expressed in d pi were in the following range (values in parentheses are the mean): 1 d pi, 21–25 h pi (23.0 h pi); 2 d pi, 43–55 h pi (46.2 h pi); 3 d pi, 69–77 h pi (71.1 h pi); and 4 d pi, 92–103 h pi (96.0 h pi).

## Plaque assay

For virus titration, a plaque assay was performed on Vero cells according to the manual provided on the website of the National Institute of Infectious Diseases [23] with modifications. In brief, cell monolayers grown in 12-well plates (Corning Costar®; No. 3513) were inoculated with virus samples in 0.4 ml per well of culture medium with reduced heat-inactivated FBS (2%) and were incubated at 37˚C for 2 h. After removal of the samples, the cells were incubated in 2 ml of overlay medium consisting of EMEM (Nissui Pharmaceuticals, Tokyo, Japan; No.05901), 0.22% (w/v) NaHCO₃, 2% (v/v) heat-inactivated FBS, 2 mM L-glutamine (Nacalai Tesque; No. 16948–04), and 1% (w/v) methylcellulose (Tokyo Chemical Industry, Tokyo, Japan; No. M0185) at 37˚C for 5 days. The cells were fixed with 10% (v/v) neutral buffered formalin (Wako Pure Chemical Industries; No. 133–10311) and stained with 0.0375% (w/v) methylene blue. Wells with 10–100 plaques were used to calculate the virus titer in plaque formation units (PFU). For the comparative plaque assay, cells were similarly grown and infected,

but overlaid with 2.5 ml per well of DMEM (Nissui Pharmaceuticals; No. 05919) supplemented with 0.2% (w/v) NaHCO$_3$, 2% (v/v) heat-inactivated FBS, 2 mM L-alanyl-L-glutamine (Nacalai Tesque; No. 04260–64), 0.1 mM nonessential amino acids, and 1.25% (w/v) methylcellulose. JEV plaque formation in Vero cells was not so different between the DMEM overlay medium and the EMEM overlay medium (S1B Fig). After fixation and staining, plates were scanned with a document scanner, and image data were obtained (S11 Fig). To improve plaque visibility, an invert filter was applied to the image data, and highlights were adjusted using Photoshop Elements (version 14; Adobe Systems, San Jose, CA, USA).

## RNA purification

Total RNA was extracted and purified from cells and culture supernatant using the Blood/Cultured Cell Total RNA Mini Kit (Favorgen Biotech, Pingtung City, Taiwan; No. FABRK 001–2) and the Viral Nucleic Acid Extraction Kit I (Favorgen Biotech; No. FAVNK 001–2), respectively, according to the manufacturer's instructions.

## Quantification of RNA by quantitative reverse transcriptase-polymerase chain reaction (qRT-PCR)

To determine the copy numbers of JEV RNA (Nakayama strain; GenBank accession No. EF571853.1), the RNA fraction was subjected to one-step qRT-PCR using the THUNDER-BIRD® Probe One-step qRT-PCR Kit (Toyobo, Osaka, Japan; No. QRZ-101) with primers #1 and 2 and a hydrolysis probe #3 (Table 1). The following reactions were performed on the LightCycler® 96 system (Roche Applied Science): Reverse transcription at 50˚C for 10 min and denaturation at 95˚C for 1 min, followed by 40 cycles of 95˚C for 15 s and 56˚C for 45 s. To normalize cellular JEV RNA levels, the copy numbers of mRNA encoding glyceraldehyde-3-phosphate dehydrogenase (GAPDH) (human, GenBank accession No. NM_002046.3; *Chlorocebus sabaeus*, GenBank accession No. XM_007967342) in the same RNA samples were determined with primers #6 and 7 and a hydrolysis probe #8 (Table 1; the sequences of #6−8 are homologous to both the human and monkey sequences) under the same qRT-PCR conditions, except that the annealing temperature was 60˚C.

A standard curve was established using serial dilutions of plasmids encoding the JEV E protein (bases 962 to 2491) or human GAPDH. For construction of the plasmid encoding the JEV E protein, the RNA fraction prepared from a JEV (Nakayama strain) stock was reverse transcribed with primer #4 (Table 1) using the PrimeScript™ II 1st Strand cDNA Synthesis Kit (Takara Bio, Shiga, Japan; No. 6210A). The first strand cDNA was amplified on a PCR Thermal Cycler Dice (Takara Bio; also used for all the PCR reactions below) using PrimeSTAR® MAX DNA polymerase (Takara Bio; No. R045A) and primers #4 and 5 (Table 1) under the following conditions: 35 cycles of 98˚C for 10 s, 56˚C for 5 s, and 72˚C for 90 s. The amplicon was cloned into a T-vector pMD20 (Takara Bio; No. 3270) using the Mighty TA-cloning Reagent Set for PrimeSTAR® (Takara Bio; No. 6019). For construction of the plasmid encoding human GAPDH, the total RNA fraction from Huh7.5.1–8 cells was reverse transcribed with an oligo-dT primer using the Transcriptor First Strand cDNA Synthesis Kit (Roche Applied Science, Penzberg, Upper Bavaria, Germany; No. 04379012001). *GAPDH* cDNA was obtained by PCR using primers #9 and 10 (Table 1) at 35 cycles of 98˚C for 10 s, 58˚C for 5 s, and 72˚C for 60 s and cloned into the EcoRI−BglII sites of plasmid pBSnFLcHA [24]. Both constructs were purified by the PureYield™ Plasmid Miniprep System (Promega, Madison, WI, USA; No. A1222), and their sequences were verified by Sanger sequencing.

**Table 1. Primers and probes used in this study.**

| Target and GenBank accession No. | # | Name | Sequence (base positions in the reference sequence) | Used for: |
|---|---|---|---|---|
| *JEV strain Nakayama* EF571853.1 | 1 | JE3-1357-1379-F | 5'-GGAGAACAATCCAGCCAGAAAAC-3'<br>(bases 1357 to 1379) | qRT-PCR |
| | 2 | JE3en-1470-1494-R | 5'-CATTGGGTGTTACTGTAAACTTTGC-3'<br>(complementary to bases 1470 to 1494) | qRT-PCR |
| | 3 | JE3-1427-1457-Taq | 5'-[FAM]-AAACCATGGGAATTATTCAGCGCAAGTTGGG-[TAMRA]-3'<br>(bases 1427 to 1457) | qRT-PCR |
| | 4 | JE2491-R23 | 5'-GCACATCCAGTGTCAGCATGCAC -3'<br>(complementary to bases 2469 to 2491) | cDNA amplification |
| | 5 | JE962-F24 | 5'-CGCTCCGGCTTACAGTTTCAACTG-3'<br>(bases 962 to 985) | cDNA amplification |
| Human *GAPDH* NM_002046.3 | 6 | Cs_GAPDH-F | 5'-AGGTCGGAGTCAACGGATTTG-3'<br>(bases 116 to 136) | qRT-PCR |
| | 7 | Cs_GAPDH-R | 5'-GGTCATTGATGGCAACAATATCCA-3'<br>(complementary to bases 195 to 208)- | qRT-PCR |
| | 8 | Cs_GAPDH-P | 5'-[FAM]-TGGTCACCAGGGCTGCTTTAACTCTGG-[TAMRA]-3'<br>(bases 152 to 179) | qRT-PCR |
| | 9 | GAPDH-HA-IF1 | 5'-TGGCGGCCGCGAATTCACCATGGGGAAGGTGAAG-3'<br>(bases 100 to 133) | cDNA amplification |
| | 10 | GAPDH-HA-IF2 | 5'-CATATGGGTAAGATCTCTCCTTGGAGGCCATGTG-3'<br>(complementary to bases 1074 to 1107) | cDNA amplification |
| Human *RIG-I* AF038963.1 | 11 | Sumpter F | 5'-GTCCGGCCTCATTTCCTCGGAAAATC-3'<br>(bases 34 to 69) | cDNA amplification [16] |
| | 12 | Sumpter R | 5'-GGTACAAGCGATCCATGATTATACCCACTATGTTTG-3'<br>(complementary to bases 2992 to 3027) | cDNA amplification [16] |
| | 13 | 216_F21 | 5'-TGGACCCTACCTACATCCTGA-3'<br>(bases 216 to 236) | Sequencing |
| | 14 | 439_R21 | 5'-GAAATCCCAACTTTCAATGGC-3'<br>(complementary to bases 419 to 439) | Sequencing |
| | 15 | 601_F23 | 5'-GATGGCAGGTGCAGAGAAATTGG-3'<br>(bases 601 to 623) | Sequencing |
| | 16 | 1390_F23 | 5'-GGTTGGTGTTGGGGATGCCA-3'<br>(bases 1390 to 1412) | Sequencing |
| | 17 | 160_R21 | 5'-GTGTCCCTCATCAGCTGAGC-3'<br>(complementary to bases 1582 to 1602) | Sequencing |
| | 18 | 1978_R20 | 5'-CTCATTGCTGGGATCCCTGG-3'<br>(complementary to bases 1959 to 1978) | Sequencing |
| | 19 | 2129_F22 | 5'-CCTGGCATATTGACTGGACGTG-3'<br>(bases 2129 to 2150) | Sequencing |
| | 20 | 2989_R22 | 5'-TCTTCTCCACTCAAAGTTACTC-3'<br>(complementary to bases 2968 to 2989) | Sequencing |

## Cloning and sequencing of RIG-I (DDX58) cDNA

Total RNA from cells was reverse transcribed with an oligo dT primer using the PrimeScript™ II 1st Strand cDNA Synthesis Kit (described above). The first strand cDNA was used as a template to amplify a cDNA fragment encoding RIG-I (DDX58) (GenBank accession No.

AF038963.1; bases 34 to 3027) by PCR using primers #11 and 12 (Table 1) and PrimeSTAR® GXL DNA polymerase (Takara Bio; No. R050A) at 35 cycles of 98˚C for 10 s, 60˚C for 15 s, and 68˚C for 3 min. The resultant amplicons were separated on a 0.7% agarose gel, purified using the Gel/PCR Extraction Kit (FastGene, NIPPON genetics, Tokyo, Japan; No. FG-91302), and then directly sequenced using primers #13 and 14 (Table 1). The amplicons were cloned into pCR®Blunt II-TOPO® (Thermo Fisher Scientific, Waltham, MA, USA; No. K2800-20), purified, and sequenced with primers #11–20 (Table 1). Upon sequence determination, nucleotide variations were excluded that were not consistent with the RNA-seq data from Huh7.5.1 (DRR018792) and Huh7.5.1–8 (DRR018793) [20] or the whole-genome sequence data from HuH-7 and Huh7.5.1–8 (DDBJ DRA database under the project ID PRJDB7928) [25].

## Other methods

All the experiments were done with three biological replicates and repeated as described in figure legends. Cell viability was determined using the Cell Proliferation Kit II XTT (Roche Applied Science; No. 11 465 015 001) according to the manufacturer's instructions. Statistical analysis was performed using an online calculator, Quick Calcs, provided on the website of GraphPad (https://www.graphpad.com/quickcalcs/), or GraphPad Prism version 7.03 (Graph-Pad Software, San Diego, CA, USA).

## Results

### Comparison of JEV production between Huh7.5.1–8, Huh7.5.1, and HuH-7 cells

First, the Huh7.5.1–8 cell line was compared with the parental Huh7.5.1 and the ancestral HuH-7 cell lines regarding JEV production. Cells were infected with JEV (Nakayama strain, unless otherwise noted), and then the amount of infectious virus particles (titer) in culture supernatant and JEV-induced cell death were monitored during 1–4 d pi. The relative JEV titer of these cell lines increased and peaked at 2–3 d pi in a similar manner (Fig 1A). Under the same seeding conditions, doubling times of Huh7.5.1–8, Huh7.5.1, and HuH-7 cells were not significantly different (after Bonferroni correction, p > 0.0167) (S2B Fig). Thus, JEV production appeared to be comparable between the three cell lines and not enhanced in this lineage unlike for HCV production. In a parallel cell viability assay (Fig 1B), susceptibility to JEV-induced cell death was not greatly different between the three cell lines. Next, these cell lines were infected with JEV, overlaid with methylcellulose-containing medium, and then compared regarding JEV plaque formation. Huh7.5.1–8 and Huh7.5.1 cells similarly developed JEV plaques, whereas HuH-7 cells began to die at 3 d pi and detached from the surface of the well irrespective of infection at 5 d pi (Fig 1C). These results indicated that the Huh7.5.1–8 and Huh7.5.1 cell lines are better than the HuH-7 cell line for a JEV plaque assay. Although Huh7.5.1–8 and Huh7.5.1 cell lines were comparable in terms of JEV infection, the Huh7.5.1–8 cell line has the advantage of being phenotypically more stable than the Huh7.5.1 cell line [20]. Thus, we hereafter focused on the Huh7.5.1–8 cell line and compared it with the Vero cell line regarding JEV production.

### JEV production in Huh7.5.1–8 and Vero cell lines

For comparative analysis, we used the Vero cell subline JCRB9013, in which JEV production was almost equivalent to or slightly higher than that in other Vero cell sublines (S4 Fig). Huh7.5.1–8 and Vero cells under high and low confluency were infected with JEV at MOI 0.1, and then the virus titer in culture supernatant was monitored from 1 to 4 d pi (Fig 2). The

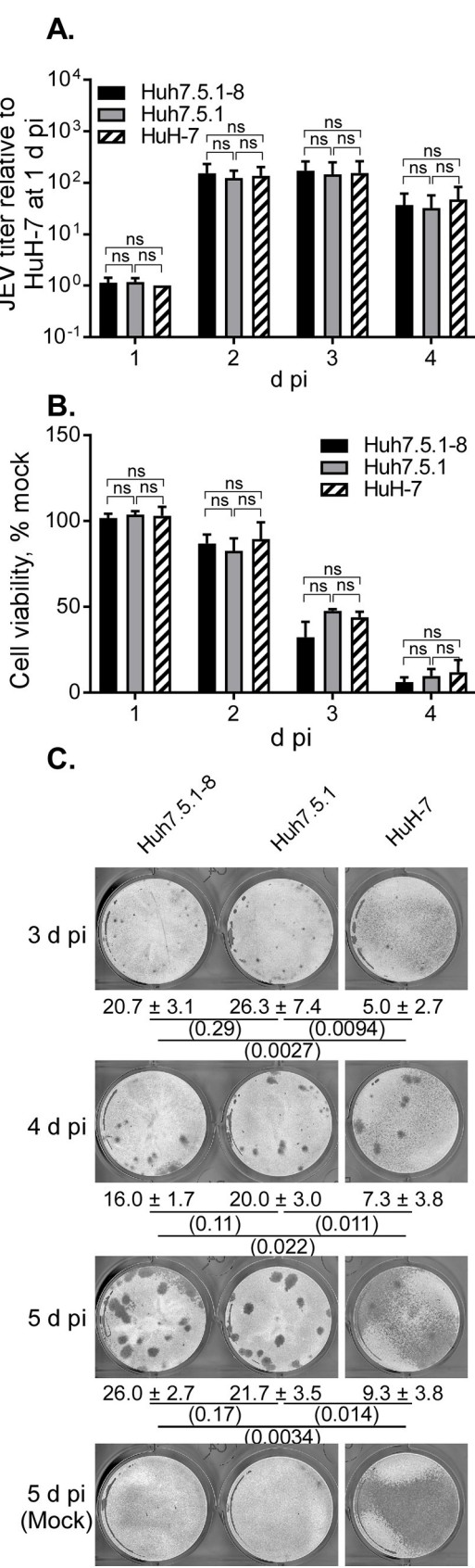

**Fig 1. Comparison of the course of JEV infection between Huh7.5.1–8, Huh7.5.1, and HuH-7 cells.** (A, B) Cells were seeded at $1 \times 10^5$ cells per well of a 24-well plate one day before infection and then infected with JEV at MOI 0.1. Culture supernatants were harvested at the indicated times to determine virus titers in PFU per well. Panel A shows the relative JEV titer calculated by dividing each titer value by that of HuH-7 cells at 1 d pi from the same experiment (actual value, $[4.90-10.5] \times 10^5$ PFU/well). Panel B shows cell viability expressed as a % of that of mock-infected cells. For both panels, bars with error bars represent the mean ± standard deviation (SD) of three independent experiments. Statistical significance between cell lines was determined by a one-sample t test (for comparison with a hypothetical value,1) or an unpaired two-tailed t test using Bonferroni correction and p values less than 0.0167 were considered statistically significant. ns, not significant. (C) Cells were seeded at $4 \times 10^5$ cells per well of a 12-well plate one day before infection and then infected with JEV at 40 PFU (determined with Vero cells) per well. The cells were fixed and stained at 3−5 d pi. Values given below the images are plaque numbers expressed as the mean ± SD of triplicates from one representative experiment. Statistical significance was determined as described above. Values in parentheses are p values, and those less than 0.0167 were considered statistically significant. Similar results were obtained in two other independent experiments (S3 Fig).

culture supernatant of Huh7.5.1−8 cells exhibited a higher relative virus titer than that of Vero cells, particularly during the early infection times. The difference in the relative virus titer was remarkable in high-cell-density infection (Fig 2A) compared with low-cell-density infection (Fig 2B), implying that cell-cell contacts promote viral spread in Huh7.5.1−8 cells. The difference in the titer was not accounted for by the growth rates of the two cell lines, because their doubling times under high confluency conditions were nearly equal (S2 Fig). A parallel viability assay showed that Huh7.5.1−8 cells lost viability faster than Vero cells irrespective of cell density (Fig 3). The same trends were observed in JEV production and cell viability at smaller MOI (0.01), although high confluency appeared to be needed (S5 Fig). These results showed that JEV production in the Huh7.5.1−8 cell line is higher than that in the Vero cell line during early infection times and that the Huh7.5.1−8 cell line is more susceptible to JEV-induced cell death than the Vero cell line.

## Particle-to-PFU ratio, replication kinetics, and virus release

The particle-to-PFU ratio was calculated for Huh7.5.1-8- and Vero-derived JEV particles, and the ratio of the values of both particles was determined (Fig 4). Both virus particles showed a similar particle-to-PFU ratio, ruling out the possibility that Huh7.5.1-8-derived virus particles had a higher infectivity than Vero-derived ones. The reason for large variations found at 2 and 3 d pi remains unknown, but that at 3 d pi may have been partly due to variable amounts of non-infectious viral RNA leaking from dead Huh7.5.1−8 cells (Fig 3). In a representative experiment (S6 Fig), the actual value of the particle-to-PFU ratio was minimally ~600, which is roughly consistent with a previous report [26]. Taken together with the results of Fig 2, these results suggested that Huh7.5.1−8 cells produce a higher amount of JEV particles than Vero cells during early infection times.

Next, we compared the JEV replication kinetics between Huh7.5.1−8 and Vero cells by monitoring intracellular and extracellular viral RNA levels during 0−73 h pi. An initial rise in intracellular viral RNA level in Huh7.5.1−8 cells was detected as early as 9 h pi, whereas the rise for Vero cells was not remarkable during 0−12 h pi (Fig 5A [a]). Similarly, the level of extracellular viral RNA from Huh7.5.1−8 cells rose earlier than that from Vero cells (Fig 5B [a]). Although the level of intracellular viral RNA was not different between the two cell lines after 12 h pi (Fig 5A [b]), the level of extracellular viral RNA was higher for Huh7.5.1−8 cells than for Vero cells during 12−48 h pi (Fig 5B [b]). Furthermore, the relative amount of extracellular viral RNA expressed as a percentage of the total (extracellular plus intracellular) amount of viral RNA was about 2-fold higher in Huh7.5.1−8 cells than in Vero cells at 1 d pi (Fig 6). Thereafter, the value for Huh7.5.1−8 cells varied with each experiment, possibly due to variation in virus-induced cell death, but leading to no difference between the values for

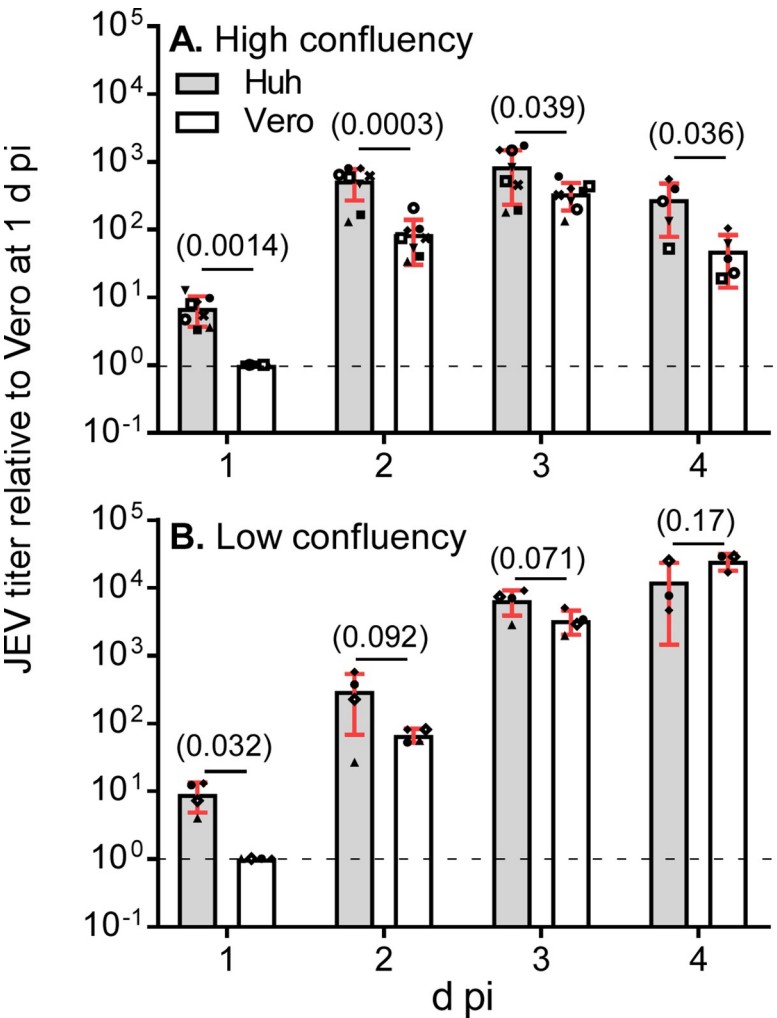

**Fig 2. JEV titer of Huh7.5.1–8 cells was higher than that of Vero cells early in infection.** (A, B) Cells were seeded at $1 \times 10^5$ (A; high confluency conditions) or $2 \times 10^4$ (B; low confluency conditions) cells per well of a 24-well plate one day before infection and then infected with JEV at MOI 0.1. Culture supernatants were harvested at 1−4 d pi, and virus titers in the supernatants were determined by plaque assay. Each symbol represents the relative titer value calculated by dividing each titer value in PFU/well by that of Vero cells at 1 d pi in the same experiment. Bars with error bars represent the mean ± SD of the relative values from multiple independent experiments. Because the 4 d pi time point was omitted in several experiments, numbers of experiments differ at 1−3 d pi and at 4 d pi: A, n = 8 [1−3 d pi] or 5 [4 d pi]; B, n = 5 [1−3 d pi] or 3 [4 d pi]. Statistical significance was determined by a one-sample t test (1 d pi) or unpaired two-tailed t test (2−4 d pi). Values in parentheses indicate p values, and those less than 0.05 were considered statistically significant. Actual titers in PFU/well for Vero cells at 1 d pi were $(8.75–24.6) \times 10^4$ (A) and $(2.73–10.2) \times 10^3$ (B). Huh, Huh7.5.1–8 cells.

Huh7.5.1–8 and Vero cells. Thus, Huh7.5.1–8 cells may release JEV more efficiently than Vero cells at least at 1 d pi. Collectively, these results suggested that rapid viral replication kinetics and efficient early virus release contribute to high JEV production in Huh7.5.1–8 cells.

## JEV plaque formation

JEV plaque formation (size and number) were compared between Huh7.5.1–8 and Vero cell lines at 3−5 d pi. As shown in Fig 7A and S7A Fig, clearly visible plaques were developed in Huh7.5.1–8 cells at as early as 3 d pi, but not in Vero cells at that time. In the course of the

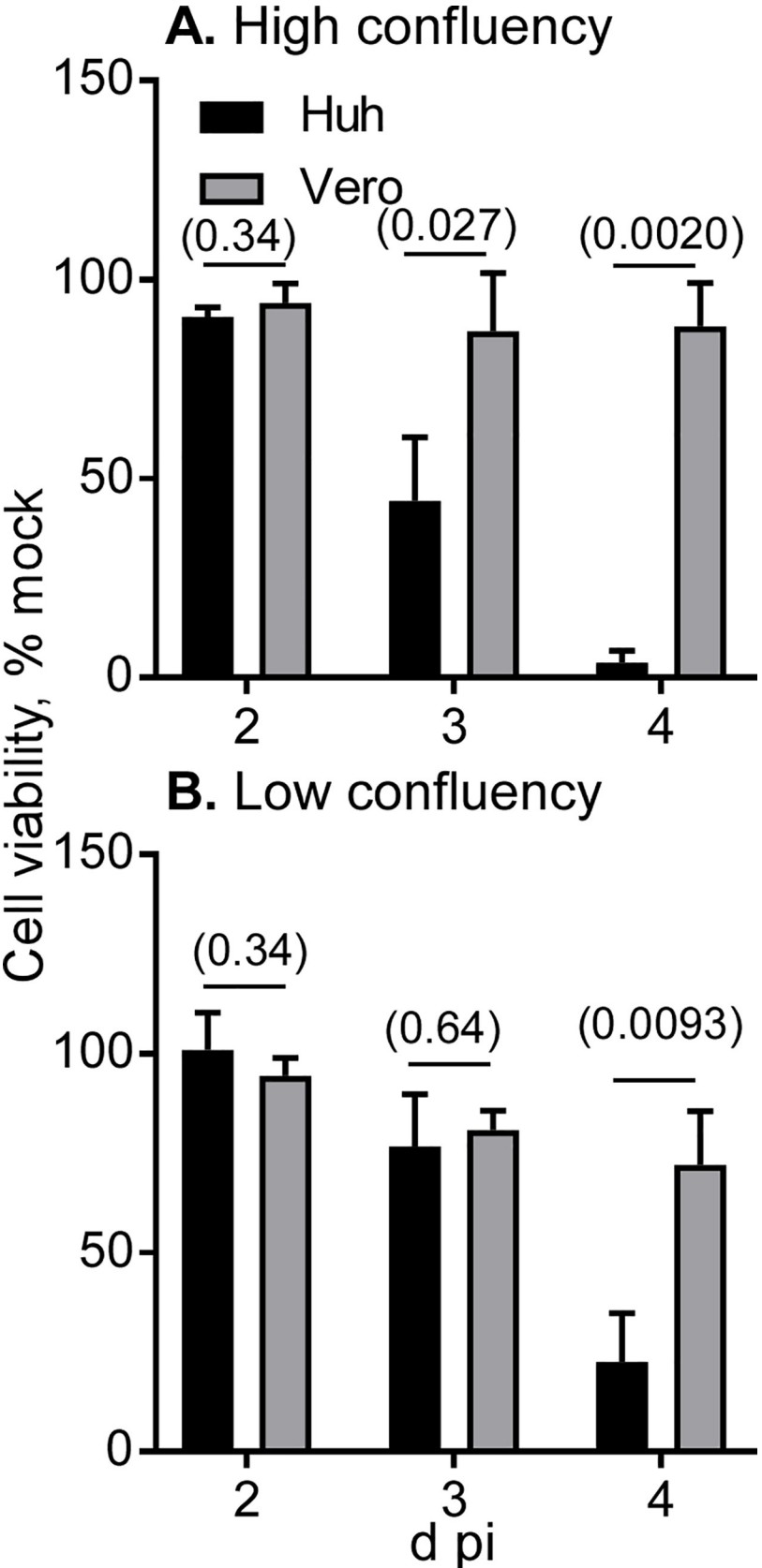

**Fig 3. Huh7.5.1–8 cells were more susceptible to JEV-induced cell death than Vero cells.** Cells were seeded and infected with JEV at MOI 0.1 under high (A) and low (B) confluency conditions as described in the legend to Fig 2. Cell viability was determined at 2−4 d pi and expressed as % of viability of mock-infected cells. Bars with error bars represent the mean ± SD of three independent experiments. Statistical significance between Huh7.5.1–8 cells (Huh) and Vero cells was determined by an unpaired two-tailed t test. Values in parentheses are p values, and those less than 0.05 were considered statistically significant.

infection, plaques in Huh7.5.1–8 cells grew more rapidly than those in Vero cells. Similar results were observed when the cells were infected with another JEV strain, rAT (Fig 7B and S7B Fig). These results showed that the Huh7.5.1–8 cell line supports more rapid JEV plaque formation than the Vero cell line. The rapid plaque formation in Huh7.5.1–8 cells appears to reflect their high JEV productivity and susceptibility to JEV-induced cell death. The plaque numbers of the Nakayama strain at 5 d pi were comparable between the two cell lines (Fig 7A and S7A Fig), whereas those of the rAT strain tended to be higher in the Huh7.5.1–8 cell line than in the Vero cell line (Fig 7B and S7B Fig). These results indicated that the difference in plaque numbers between the two cell lines varies by each virus strain.

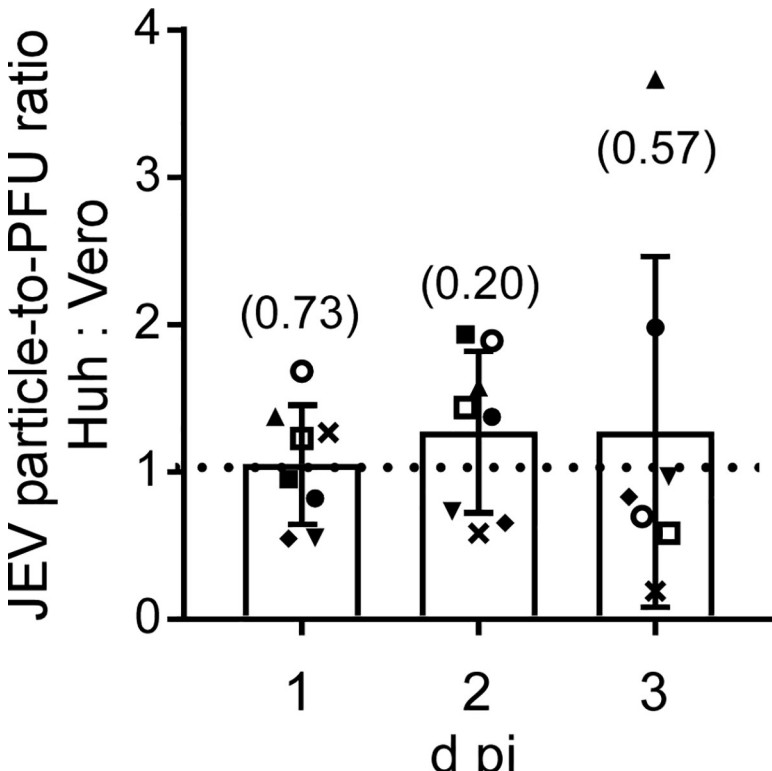

**Fig 4. Particle-to-PFU ratio was comparable between Huh7.5.1-8-derived and Vero-derived JEV.** Cells were infected with JEV at MOI 0.1 under high confluency conditions as described in the legend to Fig 2, and the culture supernatants were harvested at the indicated times. RNA was extracted and purified from each culture supernatant, and viral RNA copy number in the supernatant was determined by qRT-PCR and regarded as the number of virus particles. A virus titer in PFU of each culture supernatant was also determined by plaque assay, and particle-to-PFU ratio was calculated. Each symbol represents the relative value calculated by dividing the value of Huh7.5.1-8-derived JEV by that of Vero-derived JEV obtained from the same experiment. Bars with error bars represent the mean ± SD of the ratio from eight (1 and 2 d pi) or seven (3 d pi) independent experiments (the reason for the different sample sizes was that one experiment lacked data for 3 d pi). Statistical significance was determined by a one-sample t test. Values in parentheses indicate p values, and those less than 0.05 were considered statistically significant.

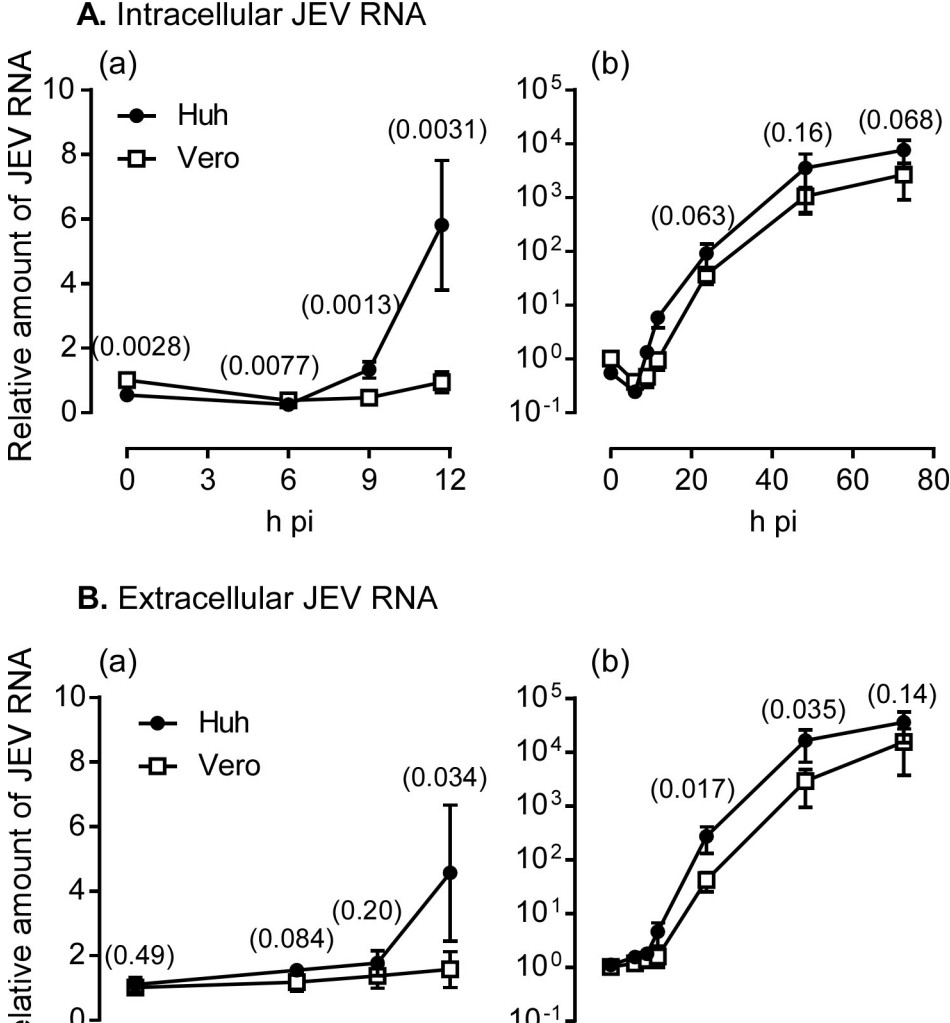

**Fig 5. Huh7.5.1–8 cells exhibited more rapid viral replication kinetics than Vero cells.** Cells were infected with JEV at MOI 0.1 under high confluency conditions as described in the legend to Fig 2. Cells and culture supernatants were harvested at 0, 6, 9, 12, 24, 48, and 73 h pi, and RNA was purified. Viral RNA copy number in each RNA fraction was determined by qRT-PCR. The copy number of intracellular viral RNA was normalized by that of *GAPDH* RNA in the same fraction, whereas the amount of extracellular viral RNA was expressed as the total copy number in the culture supernatant of each well. The relative amounts of intracellular (A) and extracellular (B) viral RNA were calculated by dividing each value by that of Vero cells at 0 h pi in the same experiment. Data represent the mean ± SD of the relative values from four independent experiments. Graph (a) is an enlarged view of 0–12 h pi drawn on a linear scale, whereas graph (b) is a whole view drawn on a log scale. Some error bars are not visible due to their small size. Statistical significance was determined by a one-sample t test (0 h pi) or an unpaired two-tailed t test (other time points). Values in parentheses indicate p values, and those less than 0.05 were considered statistically significant. Huh, Huh7.5.1–8 cells.

## YFV production

To compare the production of another flavivirus, Huh7.5.1–8 and Vero cells were infected with the YFV vaccine strain (17D-204), and then the virus titer in culture supernatant and YFV-induced cell death were monitored under high and low confluency during 1–4 d pi. The

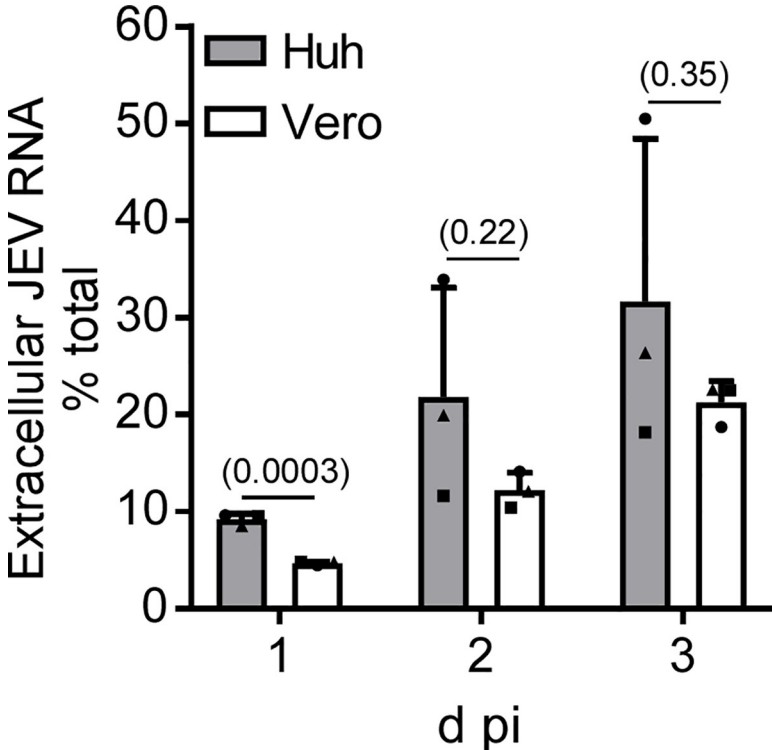

**Fig 6. Huh7.5.1–8 cells released JEV more efficiently than Vero cells early in infection.** Cells were infected with JEV at MOI 0.1 under high confluency conditions as described in the legend to Fig 2. Cells and culture supernatant were harvested at the indicated times. RNA was extracted and purified, and the viral RNA copy number in each RNA fraction was determined by qRT-PCR. The extracellular amount of viral RNA was expressed as a percentage of the total (extracellular plus intracellular) amount of viral RNA. Data represent the mean ± SD of three independent experiments. Each symbol represents the mean from one experiment. Statistical significance was determined by an unpaired two-tailed t test. Values in parentheses indicate p values, and those less than 0.05 were considered statistically significant.

culture supernatant of Huh7.5.1–8 cells exhibited a higher YFV titer than that of Vero cells during early infection times, as shown by three independent experiments (Fig 8A and 8B and S8 Fig). Because variations between experiments performed on different days were too large, the YFV titer ratio of Huh7.5.1–8 cells to Vero cells was calculated on each time point, and the mean of the three experiments was taken (Fig 8C and 8D). The combined data showed that virus yield in Huh7.5.1–8 cells was significantly higher than that in Vero cells for high confluency at 2 d pi and low confluency at 1 and 2 d pi. Furthermore, Huh7.5.1–8 cells were more susceptible to YFV-induced cell death than Vero cells (Fig 8E and 8F), as observed with JEV. These results suggested that the Huh7.5.1–8 cell line, compared with the Vero cell line, has a higher virus productivity and susceptibility to virus-induced cell death upon flavivirus infection. Next, we compared the YFV plaque formation between the two cell lines (Fig 9 and S9 Fig). YFV plaques began to appear at 4 d pi in both cell lines. Although plaques in Huh7.5.1–8 cells were slightly smaller than those in Vero cells, Huh7.5.1–8 cells developed a several fold higher number of YFV plaques than Vero cells. These results suggested that the high YFV productivity and susceptibility to YFV-induced cell death of Huh7.5.1–8 cells can be mainly attributed to their high infection efficiency. Although YFV plaques developed in Vero cells at 4 d pi in Fig 9, no cell death was observed with these cells at this time point in Fig 8E and 8F. A possible explanation for this discrepancy in cell death may be the difference in culture conditions: The serum concentration was 10% for Fig 8 and 2% for Fig 9. In addition, the medium

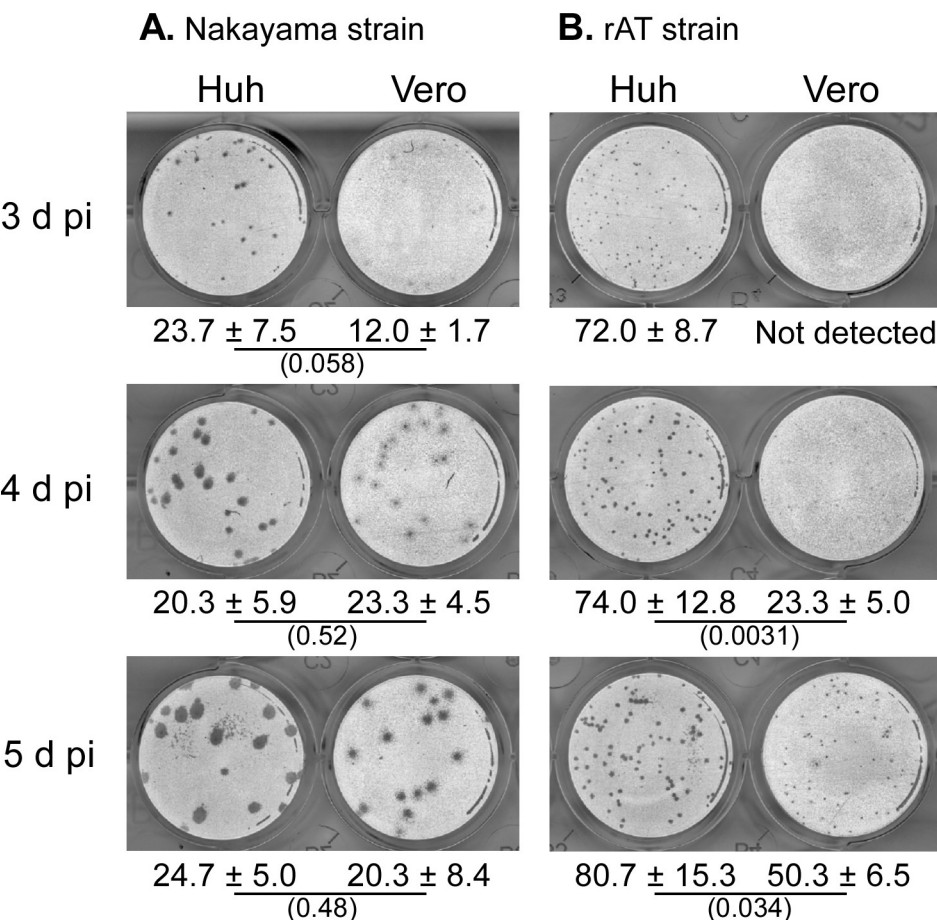

**A.** Nakayama strain

Huh    Vero

3 d pi

23.7 ± 7.5    12.0 ± 1.7
(0.058)

4 d pi

20.3 ± 5.9    23.3 ± 4.5
(0.52)

5 d pi

24.7 ± 5.0    20.3 ± 8.4
(0.48)

**B.** rAT strain

Huh    Vero

72.0 ± 8.7    Not detected

74.0 ± 12.8    23.3 ± 5.0
(0.0031)

80.7 ± 15.3    50.3 ± 6.5
(0.034)

**Fig 7. Huh7.5.1–8 cells supported a more rapid development of JEV plaques than Vero cells.** Representative black-and-white inverted images of plaques of the JEV Nakayama strain (A) and the rAT strain (B) in Huh7.5.1–8 (Huh) and Vero cells. Cells were fixed and stained at the indicated times. Values given below the images indicate plaque numbers expressed as the mean ± SD of triplicates from one representative experiment. Statistical significance was determined by an unpaired two-tailed t test. Values in parentheses indicate p values, and those less than 0.05 were considered statistically significant. Similar results were obtained in two other independent experiments (S7 Fig).

for Fig 9 contained methylcellulose. These differences may have affected Vero cell death. In the case of JEV, the lower the serum concentration is, the more cell death occurs [27].

## Confirmation of *RIG-I* sequence

Recently, codon 55 in the *RIG-I* gene from the Huh7.5.1–8 cell line was found to be heterozygous: Mutant-type ATA (Ile) and wild-type ACA (Thr) [25]. To confirm whether the Huh7.5.1–8 cell line is defective in RIG-I, we amplified the *RIG-I* cDNA derived from total RNA isolated from HuH-7, Huh7.5.1 and Huh7.5.1–8 cells. An amplicon corresponding to the expected size for the *RIG-I* sequence (~ 3 kbp) and another, with a smaller size (~ 2 kbp), were detected in the three cell lines (Fig 10A) and directly sequenced (Fig 10B). The 3-kbp and 2-kbp cDNAs from the HuH-7 cell line exclusively contained the wild-type codon 55. In contrast, the 3-kbp cDNAs from the Huh 7.5.1 and Huh7.5.1–8 cell lines contained the mutant-type codon 55, whereas the 2-kbp cDNAs from these cell lines contained the wild-type codon 55. Cloning and sequencing of the cDNAs of HuH-7 and Huh7.5.1–8 cell lines revealed that the 3-kbp cDNAs were full-length (FL1, 2), whereas the 2-kbp cDNAs lacked exons 6−12 (SV1

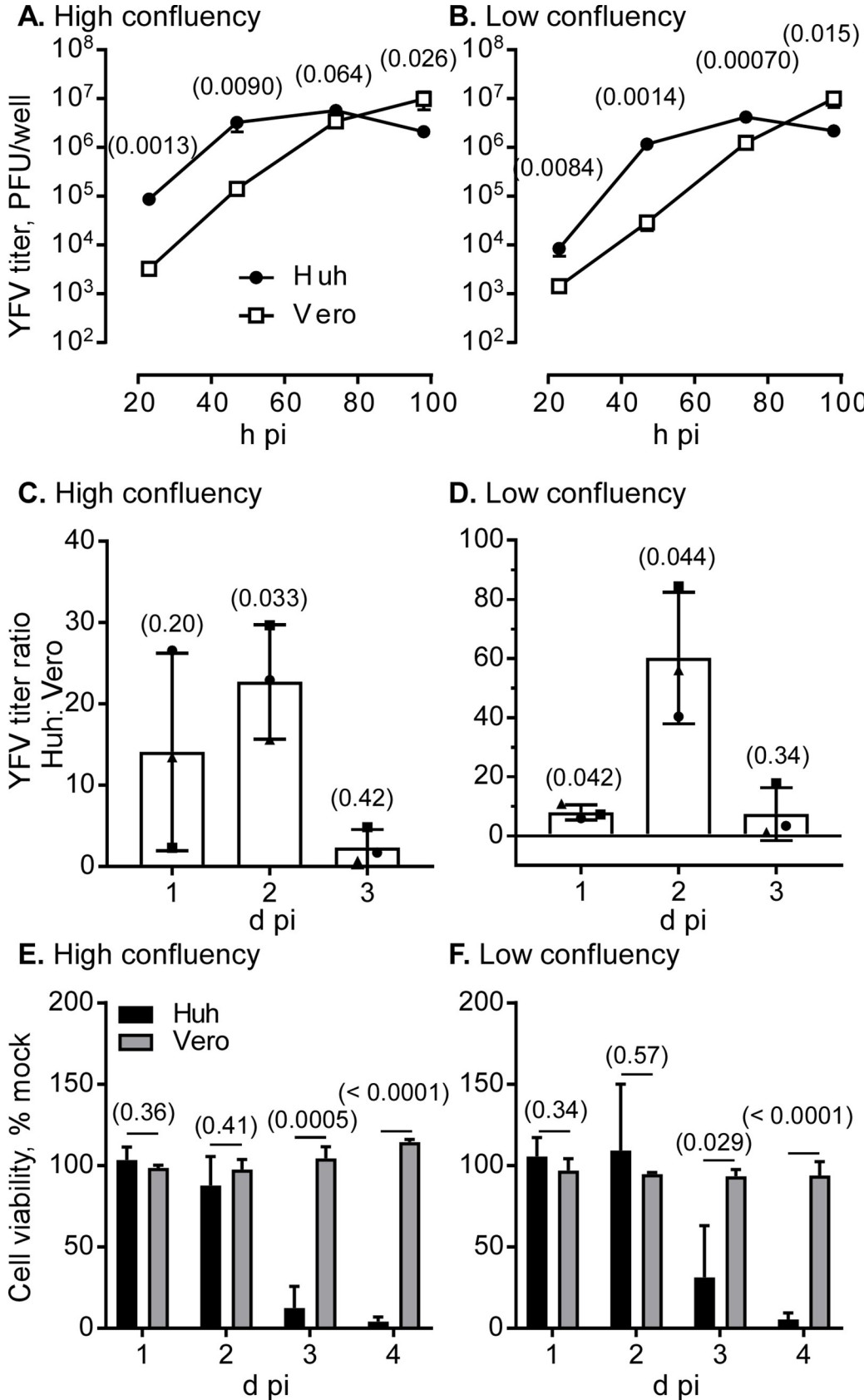

**Fig 8. Comparison of YFV production and YFV-induced cell death between Huh7.5.1–8 and Vero cells.** Cells were seeded and infected with YFV 17D-204 at MOI 0.1 under high (A, C, E) and low (B, D, F) confluency conditions as described in the legend to Fig 2. (A, B) Culture supernatants were harvested at 23, 47, 74, and 98 h pi, and virus titers in the supernatants were determined by plaque assay. Each point represents the mean ± SD of triplicates from one representative experiment. Some error bars are not visible due to their small size. Similar results were obtained in two other independent experiments (S8 Fig). (C, D) The titer ratio was calculated by dividing the titer value of Huh7.5.1–8 cells by that of Vero cells at each time point. Bars with error bars represent the mean ± SD of three independent experiments (A, B and S8 Fig). Each symbol represents the mean from one experiment. (E, F) Cell viability was determined at the indicated times. Bars with error bars represent the mean ± SD of three independent experiments. Statistical significance was determined by an unpaired two-tailed t test (A, B, E, F) or a one-sample t test (C, D). Values in parentheses indicate p values, and those less than 0.05 were considered statistically significant. Huh, Huh7.5.1–8 cells.

found in both cell lines) or 5–12 (SV2 found in Huh7.5.1–8) (Fig 10C and S10 Fig). RIG-I protein isoforms deduced from the splice variants (SV1, 2) completely lacked a helicase ATP-binding domain, indicating that they are defective like RIG-I protein with a mutated ATP-binding site [28]. These results were consistent with the above findings [25] and indicated that Huh7.5.1–8 is a RIG-I null mutant cell line.

## Discussion

Although the Huh7.5.1–8 cell line is more permissive to HCV infection than the Huh7.5.1 cell line [20], we found that this was not the case with JEV infection: The Huh7.5.1–8 and the parental Huh7.5.1 cell lines were comparable in terms of JEV productivity, susceptibility to JEV-induced cell death, and JEV plaque formation (Fig 1). Even the ancestral HuH-7 cell line was comparable except for plaque formation. These findings indicate that genetic factors contributing to the high permissiveness of the Huh7.5.1–8 cell line to HCV infection are specific rather than broadly proviral. One such genetic factor might be involved in stable HCV receptor expression [20]. Unlike the Huh7.5.1–8 and Huh7.5.1 cell lines, the HuH-7 cell line was not maintained under our culture conditions for the plaque assay (Fig 1C), likely because our HuH-7 cells had a lower fitness to cell culture than Huh7.5.1–8 and Huh7.5.1 cells.

Here, we found three differences in outputs of flavivirus infection between Huh7.5.1–8 and Vero cell lines. First, Huh7.5.1–8 cells produced higher amounts of infectious JEV and YFV than Vero cells early in infection (Figs 2 and 8A–8D). In the case of JEV, the high virus productivity of Huh7.5.1–8 cells may be attributed not to the increase in infectivity of a virus particle (Fig 4), but to viral rapid replication kinetics and efficient virus release early in infection (Figs 5 and 6). Therefore, Huh7.5.1–8 cells are more suitable than Vero cells for obtaining large amounts of flavivirus in a short period of time. Recently, HuH-7 cells were shown to produce higher amounts of Zika virus than other cells including Vero E6 cells (ATCC CRL-1586) early in infection [29]. Considering that JEV production in Huh7.5.1–8 cells and the ancestral HuH-7 cells were comparable (Fig 1A), high virus productivity early in flavivirus infection may be a common feature among the HuH-7 lineage. In addition, a similar trend was observed when HuH-7 cells were infected with Middle East respiratory syndrome coronavirus [30]. Thus, the feature may not be limited to flavivirus infection.

Second, we found that Huh7.5.1–8 cells were more susceptible to JEV/YFV-induced cell death than Vero cells (Figs 3, 8E and 8F). These characteristics of the Huh7.5.1–8 cell line may facilitate cell viability-based screenings of antiviral host factors and agents for flaviviruses. Meanwhile, the potent cell death observed in Huh7.5.1–8 cells appears to be a part of the reason for plateaued or reduced virus production late in infection. Therefore, genetic engineering of the Huh7.5.1–8 cell line to make it less susceptible to virus-induced cell death might yield a cell line with more flavivirus production.

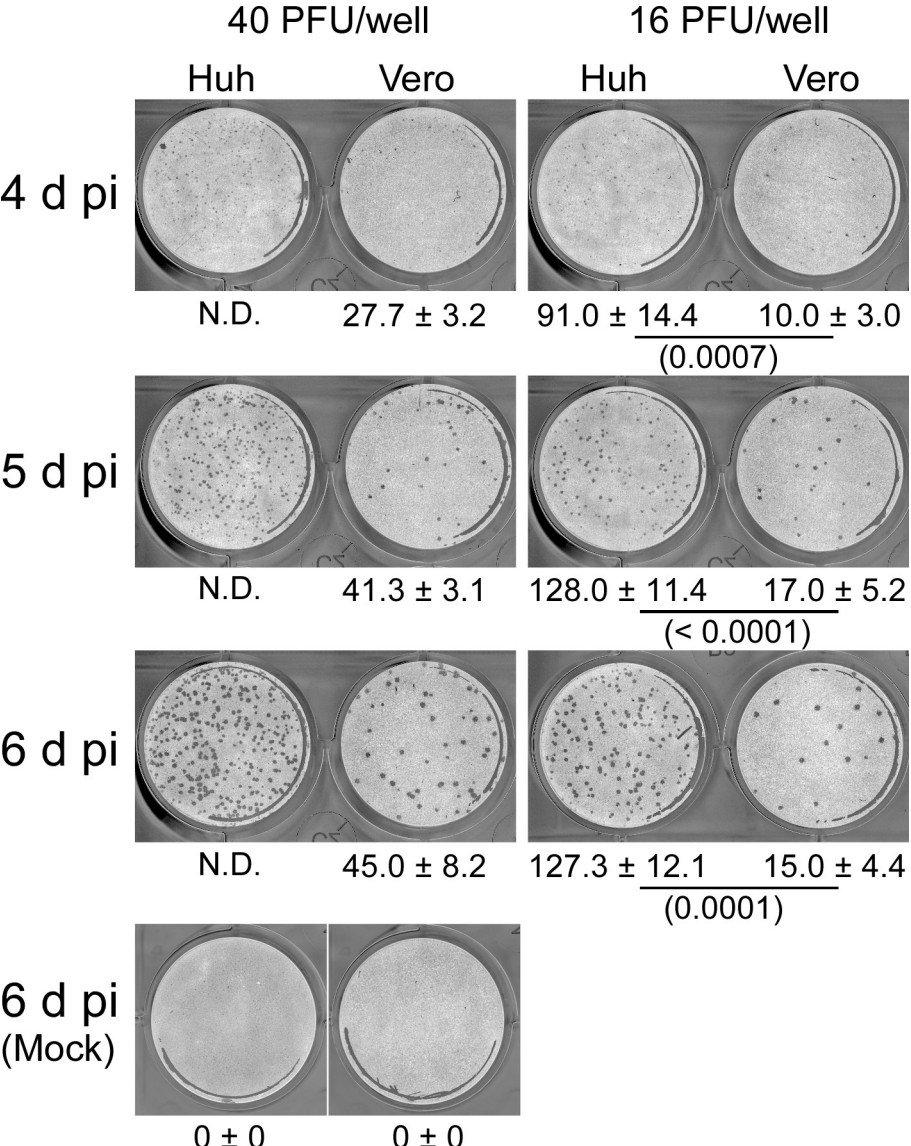

**Fig 9. Huh7.5.1–8 cells developed larger numbers of YFV plaques than Vero cells.** Representative black-and-white inverted images of YFV plaques in Huh7.5.1–8 (Huh) and Vero cells. Cells were infected with YFV (17D-204 strain) at either 40 or 16 PFU (determined with Vero cells) per well and then fixed and stained at the indicated times. Values below the images indicate plaque numbers expressed as the mean ± SD of triplicates from one representative experiment. Statistical significance was determined by an unpaired two-tailed t test. Values in parentheses indicate p values, and those less than 0.05 were considered statistically significant. Similar results were obtained in two other independent experiments (S9 Fig). N.D., not determined.

Third, we found that Huh7.5.1–8 cells developed JEV plaques rapidly (Fig 7) and formed a larger number of YFV plaques (Fig 9) compared with Vero cells, though the difference in plaque numbers between the two cell lines varied by each virus. The JEV plaque phenotype of Huh7.5.1–8 cells appears to reflect their high virus productivity and susceptibility to virus-induced cell death. Because a plaque assay is somewhat time-consuming, the plaque phenotype observed with JEV infection in Huh7.5.1–8 cells can be utilized to shorten the period of plaque assay. In addition, the high-number plaque phenotype observed with YFV infection in Huh7.5.1–8 cells can be utilized to improve the sensitivity of detection and titration of YFV by

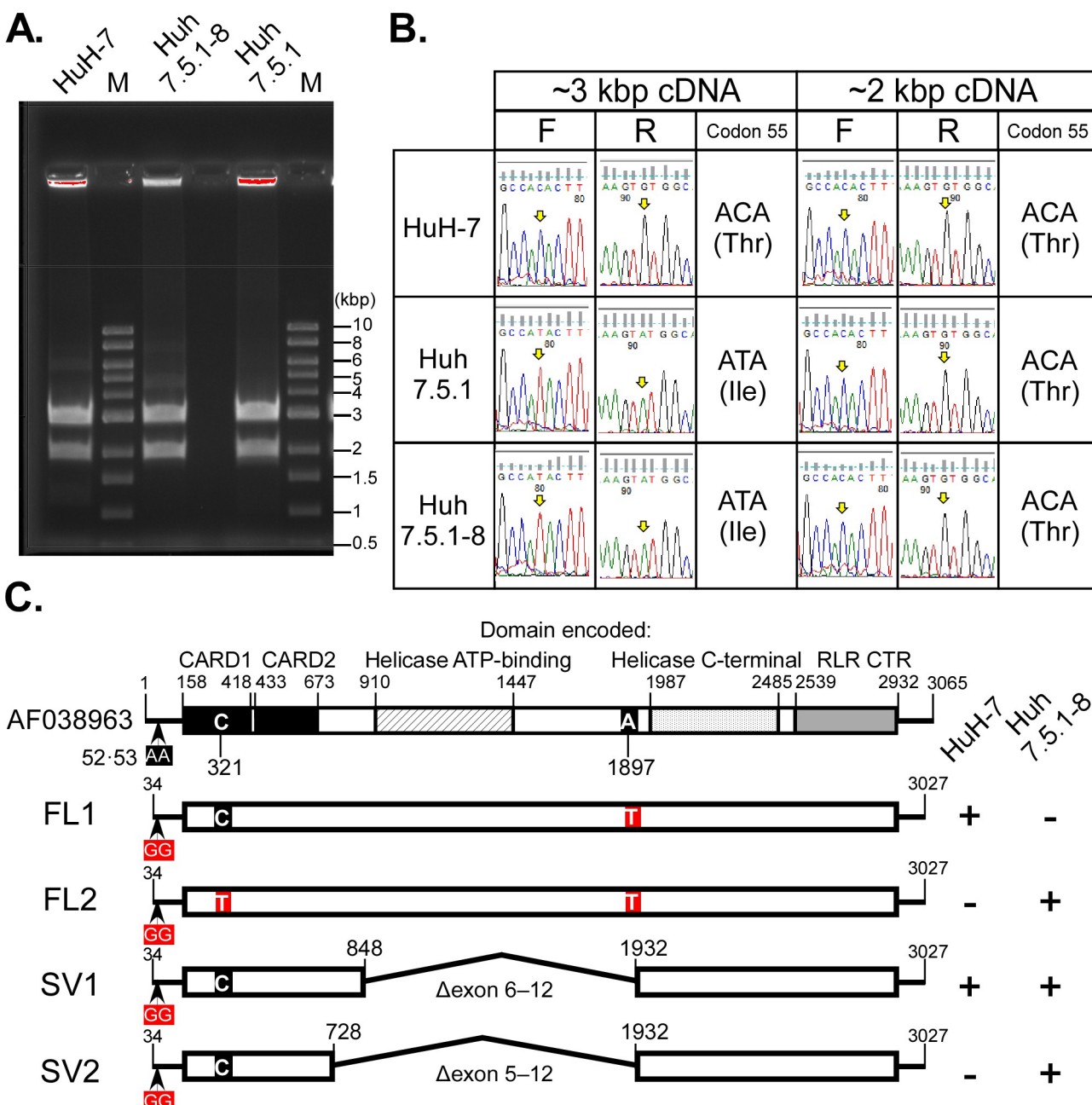

**Fig 10. Sequence analysis of *RIG-I* cDNA derived from HuH-7, Huh7.5.1, and Huh7.5.1–8 cells.** (A) Agarose gel image of RT-PCR products amplified with *RIG-I* specific primer pairs (Sumpter F and R, Table 1). M: molecular size markers. (B) Sequence chromatograms of the *RIG-I* cDNAs. Each yellow arrow indicates a nucleotide corresponding to the second base of codon 55 (base 321) of the *RIG-I* reference sequence (AF038963.1). F, forward direction; R, reverse direction. (C) Schematic representations of *RIG-I* full-length (FL) and splice variant (SV) cDNAs. The reference sequence is shown at the top. Protein domains are shown in different patterns with boundary positions: CARD, caspase activation and recruitment domain; RLR CTR, RIG-I-like receptor, C-terminal regulatory domain. Nucleotides that differed from the reference sequence are shown by red squares. The two Gs at bases 52–53 in the cDNAs were derived from the forward PCR primer used. The A/T variation at base 1897 is annotated in the single nucleotide polymorphism database (rs17217280). FL1 and SV1 sequences from HuH-7 cells were obtained from five and eight independent clones, respectively. FL2, SV1, and SV2 sequences from Huh7.5.1–8 cells were obtained from six, seven, and three independent clones, respectively. +, detected; -, not detected. The complete nucleotide sequences are shown in S10 Fig.

plaque assay. Vero cells may be more suitable to viruses requiring long-term cultivation to grow plaques, because they were more tolerant to our plaque assay conditions than Huh7.5–8 cells. In addition, the higher tolerance of Vero cells to flavivirus-induced cell death, compared with Huh7.5.1–8 cells, may be a feature that makes the Vero cell line advantageous for vaccine production.

On top of these findings, we provided evidence that Huh7.5.1–8 is a RIG-I null mutant cell line. Our results suggested that, in a Huh7.5.1–8 cell, one allele of the *RIG-I* gene generates full-length mRNA with a T55I mutation that abolishes RIG-I-mediated antiviral signaling, whereas the other allele generates defective splice variants without the mutation. One of the splice variants is also expressed in the parental HuH-7 cell line. Codon 55 of *RIG-I* gene of Huh7.5.1 cells was heterozygous as was that of Huh7.5.1–8 cells, showing that the heterozygosity of the codon was stable at least during the time span for isolation of the Huh7.5.1–8 clone from Huh7.5.1 cells. At present, the underlying mechanisms that generate splice variants and their impacts on RIG-I signaling remain to be studied. Although JEV has been restricted by RIG-I in a mouse model [31] and mouse cell lines [32, 33], JEV production and susceptibility to JEV-induced cell death were not dramatically altered in the Huh7.5.1–8 and Huh7.5.1 cell lines compared with the HuH-7 cell line (Fig 1A and 1B), implying that the RIG-I pathway may not work or may be counteracted by viral mechanisms [34] under our experimental conditions.

In conclusion, our study highlighted the characteristics of the Huh7.5.1–8 cell line that are helpful for improvement of flavivirus propagation, detection, and titration. Further study is needed to investigate the potential versatility of the Huh7.5.1–8 cell line in virus research.

## Supporting information

**S1 Fig. JEV production and plaque formation in Vero cells grown in different media.** (A) Vero cells (No. JCRB9013) were seeded at $1 \times 10^5$ cells per well of a 24-well plate one day before infection in DMEM supplemented with 10% (v/v) heat-inactivated FBS, 0.1 mM nonessential amino acids, 100 U/ml penicillin G, and 100 μg/ml streptomycin sulfate. After infection with JEV (Nakayama strain) at MOI 0.1, the cells were grown in the same medium (shown as closed circles) or EMEM-based medium as described in "Materials and methods" (shown as open squares). Culture supernatant was harvested at the indicated times, and RNA was extracted and purified to determine viral RNA copy number by qRT-PCR. Each point represents the mean ± SD of triplicates from one representative experiment. (B) Vero cells were infected with JEV and then incubated with 2.5 ml per well of DMEM overlay medium (DMEM supplemented with 0.2% [w/v] NaHCO$_3$, 2% [v/v] heat-inactivated FBS, 2 mM L-alanyl-L-glutamine, 0.1 mM nonessential amino acids, and 1.25% [w/v] methylcellulose) or EMEM overlay medium (EMEM supplemented with 0.22% [w/v] NaHCO$_3$, 2% [v/v] heat-inactivated FBS, 2 mM L-glutamine, and 1% [w/v] methylcellulose). The cells were fixed and stained at the indicated times. Values below the images are plaque numbers expressed as the mean ± SD of triplicates from one representative experiment. For both panels, statistical significance was determined by an unpaired two-tailed t test. Values in parentheses indicate p values, and those less than 0.05 were considered statistically significant. Similar results were obtained in another independent experiment. N.D., not detected.
(TIF)

**S2 Fig. Comparison of cell growth between Huh7.5.1–8, Huh7.5.1, HuH-7, and Vero cells, related to Figs 1 and 2.** (A) Cells were seeded at $1 \times 10^5$ cells per well of a 24-well plate. Cells were harvested at the indicated times and counted using the TC20 Automated Cell Counter (Bio-Rad). Results from three independent experiments are shown. (B) The doubling time of

each cell line was calculated from the slope of the above graphs from day 1 to day 3 by linear regression analysis on GraphPad Prism (ver. 7.03) and then converted to a value relative to that of Huh7.5.1–8 cells. Bar with error bars represent the mean ± SD of the relative doubling times from three independent experiments. Statistical significance was determined by a one-sample t test with Bonferroni correction. Values in parentheses are p values, and those less than 0.0167 were considered statistically significant.
(TIF)

**S3 Fig. Comparison of JEV plaque formation between Huh7.5.1–8, Huh7.5.1, and HuH-7 cells, related to Fig 1C.** Cells were seeded at $4 \times 10^5$ cells per well of a 12-well plate one day before infection and then infected with JEV at 40 PFU per well. The cells were fixed and stained at 3−5 d pi. Panels A and B represent different experiments. Values given below the images are plaque numbers expressed as the mean ± SD of triplicates from each experiment. Statistical significance was determined by an unpaired two-tailed t test with Bonferroni correction. Values in parentheses are p values, and those less than 0.0167 were considered statistically significant.
(TIF)

**S4 Fig. Comparison of JEV production among Vero cell sublines.** Vero ATCC CCL-81, Vero 76 (No. IFO50410), Vero 0111 (No. JCRB0111), and Vero C1008 (ATCC CRL-1586) cells were obtained from the National Institute of Biomedical Innovation (Osaka, Japan) as described (Sakuma C, Sekizuka T, Kuroda M, Kasai F, Saito K., Ikeda M, et al. Novel endogenous simian retroviral integrations in Vero cells: implications for quality control of a human vaccine cell substrate. Sci Rep, 2018; 8(1); 644. doi:10.1038/s41598-017-18934-2). Cells were seeded at $2 \times 10^4$ cells per well of a 24-well plate one day before infection and then infected with JEV (Nakayama strain) at MOI 0.1. Culture supernatant was harvested at the indicated times to determine virus titers by plaque assay. Each point represents the mean ± SD of triplicates from one representative experiment. Statistical significance was determined by one-way ANOVA with Dunnett's multiple-comparison post-test. *, $p < 0.05$; **, $p < 0.01$; ***, $p < 0.001$; ****, $p < 0.0001$; ns, not significant. Similar results were obtained in another independent experiment.
(TIF)

**S5 Fig. Comparison of JEV infection at MOI 0.01 between Huh7.5.1–8 and Vero cells, related to Figs 2 and 3.** Cells were seeded and infected with JEV at MOI 0.01 under high (A, C) and low (B, D) confluency conditions as described in the legend to Fig 2. (A, B) Culture supernatants were harvested at the indicated times, and virus titers in the supernatants were determined by plaque assay. Results from two independent experiments (a, b) are shown. (C, D) Cell viability was determined at the indicated times and expressed as % of viability of mock-infected cells. Bars with error bars represent the mean ± SD of three independent experiments. For all panels, statistical significance was determined by an unpaired two-tailed t test. Values in parentheses are p values, and those less than 0.05 were considered statistically significant. Huh, Huh7.5.1–8 cells.
(TIF)

**S6 Fig. Particle-to-PFU ratio of Huh7.5.1-8-derived and Vero-derived JEV, related to Fig 4.** Cells were infected with JEV at MOI 0.1 under high confluency conditions as described in the legend to Fig 2, and the culture supernatants were harvested at the indicated times. RNA was extracted and purified from each culture supernatant, and viral RNA copy number in the supernatant was determined by qRT-PCR. A virus titer in PFU of each culture supernatant was also determined by plaque assay, and particle-to-PFU ratio was calculated. Each point

represents the mean ± SD of triplicates from one representative experiment. Statistical significance was determined by an unpaired two-tailed t test. Values in parentheses indicate p values, and those less than 0.05 were considered statistically significant. Huh, Huh7.5.1–8 cells.
(TIF)

**S7 Fig. Comparison of JEV plaque development between Huh7.5.1–8 and Vero cells, related to Fig 7.** Representative black-and-white inverted images of plaques of the JEV Nakayama strain (A) and the rAT strain (B) in Huh7.5.1–8 (Huh) and Vero cells. Cells were fixed and stained at the indicated times. Values given below the images indicate plaque numbers expressed as the mean ± SD of triplicates from one representative experiment. Results from two independent experiments (a, b) are shown. Statistical significance was determined by an unpaired two-tailed t test. Values in parentheses indicate p values, and those less than 0.05 were considered statistically significant.
(TIF)

**S8 Fig. Comparison of YFV production between Huh7.5.1–8 and Vero cells, related to Fig 8.** Cells were seeded and infected with YFV at MOI 0.1 under high (a) and low (b) confluency conditions as described in the legend to Fig 2. (A, B) Culture supernatants were harvested at the indicated times, and virus titers in the supernatants were determined by plaque assay. Results from two independent experiments (A, B) are shown. Statistical significance was determined by an unpaired two-tailed t test. Values in parentheses are p values, and those less than 0.05 were considered statistically significant. Huh, Huh7.5.1–8 cells.
(TIF)

**S9 Fig. Comparison of YFV plaque development between Huh7.5.1–8 and Vero cells, related to Fig 9.** Representative black-and-white inverted images of YFV plaques in Huh7.5.1–8 (Huh) and Vero cells. Cells were infected with the indicated amount (PFU) of YFV (17D-204 strain) per well and then fixed and stained at the indicated times. Results from two independent experiments (A, B) are shown. Values below the images indicate plaque numbers expressed as the mean ± SD of triplicates from one representative experiment. Statistical significance was determined by an unpaired two-tailed t test. Values in parentheses indicate p values, and those less than 0.05 were considered statistically significant. N.D., not determined.
(TIF)

**S10 Fig. Nucleotide sequences of *RIG-I* full-length cDNAs (FL1, 2) and splice variants (SV1, 2).** Alignment of nucleotide sequences was performed using GENETYX version 14.0.0 (GENETYX Co., Tokyo, Japan).
(PDF)

**S11 Fig. Unprocessed image data, related to Figs 1C, 7, 9, S1B, S3, S7 and S9.**
(PDF)

**S1 Raw Images. Raw image data (Fig 10A).**
(PDF)

## Acknowledgments

We thank Dr. Francis V. Chisari (The Scripps Research Institute, La Jolla, CA, USA) for providing the Huh7.5.1 cell line. We also thank Dr. Tomohiko Takasaki (Department of Virology I, National Institute of Infectious Diseases) for providing the YFV 17D-204 strain.

## Author Contributions

**Conceptualization:** Kyoko Saito, Masayoshi Fukasawa, Kentaro Hanada.

**Funding acquisition:** Kyoko Saito, Masayoshi Fukasawa, Takaji Wakita, Kentaro Hanada.

**Investigation:** Kyoko Saito, Yoshitaka Shirasago, Naoki Osada.

**Methodology:** Masayoshi Fukasawa, Toshiyuki Yamaji.

**Project administration:** Kyoko Saito.

**Resources:** Ryosuke Suzuki, Takaji Wakita, Eiji Konishi.

**Supervision:** Kentaro Hanada.

**Writing – original draft:** Kyoko Saito.

**Writing – review & editing:** Kyoko Saito, Masayoshi Fukasawa, Yoshitaka Shirasago, Ryosuke Suzuki, Naoki Osada, Toshiyuki Yamaji, Takaji Wakita, Eiji Konishi, Kentaro Hanada.

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
