## [Decision Letter · Decision Letter 0]

7 Jan 2020

PONE-D-19-32355

Comparative characterization of flavivirus production in two cell lines: Human hepatoma-derived Huh7.5.1-8 and African green monkey kidney-derived Vero

PLOS ONE

Dear Dr. Saito,

Thank you for submitting your manuscript to PLOS ONE. After careful consideration, we feel that it has merit but does not fully meet PLOS ONE’s publication criteria as it currently stands. Therefore, we invite you to submit a revised version of the manuscript that addresses the points raised during the review process.

A lot of effort has been dedicated to gathering and analyzing data and the manuscript is well written and organized. However, reviewers have raised some concerns about the paper and its methods. I recommend that you consider reviewers’ recommendations and revise your paper accordingly before resubmitting your work.  I wish you the best.  

We would appreciate receiving your revised manuscript by Feb 21 2020 11:59PM. To enhance the reproducibility of your results, we recommend that if applicable you deposit your laboratory protocols in protocols.io, where a protocol can be assigned its own identifier (DOI) such that it can be cited independently in the future. For instructions see: http://journals.plos.org/plosone/s/submission-guidelines#loc-laboratory-protocols

We look forward to receiving your revised manuscript.

Kind regards,

Negin P. Martin, Ph.D.

Academic Editor

PLOS ONE

Journal Requirements:

Reviewers' comments:

Reviewer's Responses to Questions

**Comments to the Author**

1. Is the manuscript technically sound, and do the data support the conclusions?

Reviewer #1: Yes

Reviewer #2: Partly

2. Has the statistical analysis been performed appropriately and rigorously? 

Reviewer #1: No

Reviewer #2: Yes

3. Have the authors made all data underlying the findings in their manuscript fully available?

Reviewer #1: Yes

Reviewer #2: Yes

4. Is the manuscript presented in an intelligible fashion and written in standard English?

Reviewer #1: Yes

Reviewer #2: Yes

5. Review Comments to the Author

Reviewer #1: This manuscript is well-written, however several issue need to be addressed as noted below:

Comments-

• Lines 130-133- please include a brief description as to why the chosen strains for JEV and YFV were used for this study in the introduction.

• Please explain how the vaccine strains were selected and why strains SA14-14-2 and P3 or strains of that lineage weren’t included win the evaluation.

• Please clarify if Vero cells and Huh cells have the similar doubling times, and if the different Huh cell lines vary in their replication kinetics/doubling times.

• Fig 1A- Add statistics

• Fig 1B- A decrease in Huh7.5.1-8 cell viability without an increase in JEV titer is not best for vaccine production.

• “Similar results were obtained from two independent experiments…” Combine experiments and form into 1 figure.

• Combine all experiments with repeats into a single figure, wherever repeated experiments are shown as separate panels.

• Rephrase infectivity per RNA copy number to particle to PFU ratio.

• Fig 2- make all y-axis equivalent.

• Fig 2- what is the n for these experiments; ere these replicates or independent experiments -clarify

• Fig 2- Provide statistical analysis

• Fig 4- why were these time points chosen- are they relevant to the replication of JEV?

• Fig 4B- please include a description as to why there is a difference in sample size at 3 dpi

• Fig 5- Use the 2^-∆∆Ct method to demonstrate GAPDH-normalized JEV RNA-fold increases

• Fig 6- Make all of the y-axes the same to better illustrate the shift from intracellular to extracellular JEV RNA at 2 and 3 dpi. Combine all 9 data points as they are all from 3 repeats of the same experiment.

• Fig 8A- Add statistics.

• Fig 9- Please clarify how plaques are occurring at 4 dpi in Vero cells when no cell death is observed at this time-point (Figures 8C and 8D).

Reviewer #2: In the present study of “Comparative characterization of flavivirus production in two cell lines: Human hepatoma-derived Huh7.5.1-8 and African green monkey kidney-derived Vero”, Saito and colleagues compared the JEV and YFV infection in they newly developed Huh7.5.1-8 and Vero cell lines. They showed that Huh7.5.1-8 cell produced more virus particles and more susceptible to virus-induced cell death than Vero cells upon JEV and YFV infection. They further showed that this is resulted from rapid viral replication kinetics and efficient virus release but not because of higher infection efficacy. They also showed that Huh7.5.1-8 cells developed plaques more rapidly than Vero cells after JEV infection and higher numbers of plaques after YFV infection than that of Vero cells. They finally showed the Huh7.5.1-8 cell containing a functional loss of the antiviral RIG-I gene. The authors showed that Huh7.5.1-8 cells are an useful substitute cell lines for studying viruses of Flaviviridae family.

Major comments to the authors.

1. About testing the extracellular JEV RNA in Huh7.5.1-8 and Vero cell lines in figure 5B, the authors expressed the extracellular JEV RNA as the total copy number in the culture supernatant divided by that of GAPDH RNA in the whole cells from the same well. However, the authors showed that JEV infected Huh7.5.1-8 cells showed significant less cell viability in figure 3 compared with that in Vero cells. This makes the normalized GAPDH in Huh7.5.1-8 cells is much less than that in Vero cells which will increase the values in Huh7.5.1-8 cells and also affect the interpretation showed in figure 6. I think it ‘s better to compare the total RNA copy number of the extracellular RNA between two cell lines because the same cells were initially seeded and the same volume of medium was used.

2. The codon 55 of the RIG-I gene Huh-7.5.1 is heterozygous or not? How do we know the heterozygous of codon 55 in the RIG-I gene is stable during passages? The authors should discuss about this.

Minor comments to the authors.

1. For comparing the JEV production in Huh7.5.1-8 and Vero cell lines in figure 2 E and F, it seems the ratio in a bipolar distribution in 2 dpi in both high confluence and low confluence and day 1 in low confluence. Why the variation is so big?

2. For figure 2 to indicate the high confluence and low confluence, its better to labeled as high or low confluence MOI 0.01/0.1. The way the authors labeled is easy to be considered as high MOI and low MOI.

3. For comparing the infectivity of JEV in Huh7.5.1-8 and Vero cell lines in figure 4B, same question as above, the infectivity ratio in 2- and 3- dpi in a bipolar distribution which shows big variation.

6. PLOS authors have the option to publish the peer review history of their article (what does this mean?). If published, this will include your full peer review and any attached files.

Reviewer #1: No

Reviewer #2: No

---

## [Author Response · Author response to Decision Letter 0]

10 Apr 2020

RESPONSE TO EDITOR

Editor’s comment: A lot of effort has been dedicated to gathering and analyzing data and the manuscript is well written and organized. However, reviewers have raised some concerns about the paper and its methods. I recommend that you consider reviewers’ recommendations and revise your paper accordingly before resubmitting your work. I wish you the best.

Response: Thank you for your constructive feedback. We agree with you and the reviewers and have incorporated your and their suggestions throughout our manuscript. 

RESPONSES TO REVIEWER #1

 We are grateful to reviewer #1 for the critical comments and useful suggestions that have helped us to improve our paper. As indicated in the responses that follow, we have taken all these comments and suggestions into consideration in the revised version of our paper.

(Comments by reviewer #1 are shown in italics.)

Reviewer #1: This manuscript is well-written, however several issue need to be addressed as noted below:

Comment #1: Lines 130-133- please include a brief description as to why the chosen strains for JEV and YFV were used for this study in the introduction.

Response: We chose JEV (Nakayama and rAT) and YFV (17D-204) strains because these strains could be used in our biosafety level 2 laboratory. We have described the reasons in the materials and methods section (p. 6, L137�138), because we considered this section to be contextually more appropriate than the introduction. 

Comment #2: Please explain how the vaccine strains were selected and why strains SA14-14-2 and P3 or strains of that lineage weren’t included win the evaluation.

Response: The Nakayama strain was isolated in our institute in 1935 and used as a vaccine in Japan from 1954 to 1989. Given such a history, we started to use the Nakayama strain. Another reason for selecting the strain was that its plaques are easily distinguishable and of appropriate size, which is advantageous for analysis. The less virulent SA 14-14-2 strain (Yu U, Vaccine, 28, 3635, 2010; Hsieh JT et al, Nat Commun 10, 706, 2019) was not selected because of the expected small plaque morphology. For biosafety reasons, we did not examine the highly virulent strain P3 (Ni, H and Barrett AD, J Gen Virol, 77, 1449, 1996). 

Comment #3: Please clarify if Vero cells and Huh cells have the similar doubling times, and if the different Huh cell lines vary in their replication kinetics/doubling times.

Response: In the revised manuscript, we have monitored growth curves of different HuH-7-derived cells and Vero cells and have calculated their relative doubling times (S2 Fig). The new data showed that doubling times of Huh7.5.1-8, Huh7.5.1, HuH-7, and Vero cells were not significantly different. Taken together with Figs 1 and 2, this result showed that the replication kinetics per doubling time are similar between the different HuH-7-derived cell lines and that the difference of JEV production between Huh7.5.1-8 and Vero cells was not accounted for by the difference in their growth rates. In view of these results, we have added the following sentences (p.15, L263�266):

“Under the same seeding conditions, doubling times of Huh7.5.1-8, Huh7.5.1, and HuH-7 cells were not significantly different (after Bonferroni correction, p > 0.0167) (S2B Fig). Thus, JEV production appeared to be comparable between the three cell lines and not enhanced in this lineage unlike for HCV production”. 

We have also added the following sentence (p.17, L313�315): 

“The difference in the titer was not accounted for by the growth rates of the two cell lines, because their doubling times under high confluency conditions were nearly equal (S2 Fig)”. 

Comment #4: Fig 1A- Add statistics.

Response: The results of the statistical analysis have been added in the revised Fig 1A.

Comment #5: Fig 1B- A decrease in Huh7.5.1-8 cell viability without an increase in JEV titer is not best for vaccine production.

Response: We agree with the reviewer on this point. We consider that Vero cells are better than Huh7.5.1-8 cells for vaccine production and mentioned this in the original manuscript as follows (p. 29, L587�589):

 “In addition, lower susceptibility of Vero cells to flavivirus-induced cell death, compared with Huh7.5.1-8 cells, may be a feature that makes the Vero cell line advantageous for vaccine production”.

In the revised manuscript, we have changed the above sentence to

“In addition, the higher tolerance of Vero cells to flavivirus-induced cell death, compared with Huh7.5.1-8 cells, may be a feature that makes the Vero cell line advantageous for vaccine production” (p. 30, L611�613).

Comment #6: “Similar results were obtained from two independent experiments…” Combine experiments and form into 1 figure.

Response: For several sets of numeric data in the revised manuscript, we have repeated the same experiments at least three times, combined, and provided statistical analysis (Figs 1A, 1B, 3, 5, 8E, and 8F). The exception is the results of YFV titer: Because variations between experiments performed on different days were too large, individual experiments (Fig 8A, 8B, and S8 Fig) are shown as well as the combined results (Fig 8C and 8D). Thus, the sentence “similar results were obtained from two independent experiments” has been deleted from the legends of the above figures. For image data of plaque formation (Figs 1C, 7, and 9), representative experiments are shown in the main figures, and two additional experiments are shown in the supplemental figures (S3, S7 and S9 Fig). We have added this explanation to the respective legends. Representative results are still shown for two supplemental experiments which were repeated only twice (S1 and S4 Fig).

Comment #7: Combine all experiments with repeats into a single figure, wherever repeated experiments are shown as separate panels.

Response: In accordance with the reviewer’s comment, we have combined the repeated experiments, which were originally shown as separate panels, into a single figure (Figs 2 and 4). The original Fig 2E and 2F showed the Huh-to-Vero ratio of JEV titer at each time point, but did not include time-dependent changes in JEV titer. To improve this figure, we have combined the data after converting each titer value to a value relative to that of Vero cells at 1 d pi in the revised Fig 2 (similar method of data combination has been applied to Figs 1A and 5). However, representative results are still shown along with the combined results in Fig 8 (YFV titer) because of the reason mentioned in the response to comment #6. 

Comment #8: Rephrase infectivity per RNA copy number to particle to PFU ratio.

Response: The data have been converted to RNA copy number per PFU, and “infectivity per RNA copy number” has been rephrased to “particle-to-PFU ratio”.

Comment #9: Fig 2- make all y-axis equivalent.

Response: We have made all y-axes equivalent in the revised Fig 2. We have done the same in S5 and S8 Fig. 

Comment #10: Fig 2- what is the n for these experiments; ere these replicates or independent experiments –clarify

Response: The “n” for Fig 2 is the number of independent experiments. We have added this explanation in the legend.

Comment #11: Fig 2- Provide statistical analysis

Response: We have provided the results of statistical analysis in the revised Fig 2.

Comment #12: Fig 4- why were these time points chosen- are they relevant to the replication of JEV?

Response: Under our infection conditions, Huh7.5.1-8 cells started to die at 3 d pi and almost completely died at 4 d pi (Fig 3). Therefore, we chose time points (21, 45, 69, 97 h pi) roughly evenly spaced by four days. Because the original Fig 4A was a representative result partially overlapping with the combined result (Fig 4B), it has been moved to the supplemental information (S6 Fig) in accordance with comment #7.

Comment #13: Fig 4B- please include a description as to why there is a difference in sample size at 3 dpi

Response: The reason for a difference in sample size was that one out of eight experiments lacked data on 3 d pi. We have added this explanation in the legend of the revised Fig 4.

Comment #14: Fig 5- Use the 2^-∆∆Ct method to demonstrate GAPDH-normalized JEV RNA-fold increases.

Response: In the original Fig 5, we used the relative standard curve method, which is commonly used along with the 2^-∆∆Ct method for relative quantification by qPCR. More specifically, we determined the copy number by using serially diluted JEV or GAPDH cDNAs and then normalized the copy number of JEV RNA by that of GAPDH RNA. Although we think that the reviewer’s suggestion on the 2^-∆∆Ct method is valuable, the other reviewer noted that normalization of extracellular JEV RNA by cellular GAPDH RNA performed in the original Fig 5B was not appropriate because of a large difference in cell viability between Huh7.5.1-8 and Vero cells, and suggested that the amount of extracellular JEV RNA should be expressed as total copy number in the culture supernatant. Therefore, we have chosen the relative standard curve method again in the revised Fig 5. In the manual by Applied Biosystems (http://www.gu.se/digitalAssets/1125/1125331_ABI_-_Guide_Relative_Quantification_using_realtime_PCR.pdf, p.34), it is described that the relative standard curve method gives highly accurate quantitative results because unknown sample quantitative values are interpolated from the standard curve(s), implying that the relative standard curve method is not inferior to the 2^-∆∆Ct method. Furthermore, to combine four independent experiments, we have normalized the amounts of GAPDH-normalized JEV RNA (Fig 5A) and extracellular JEV RNA (copies/well) (Fig 5B) relative to that of Vero cells at 0 h pi. We have changed the y-axes of these figures to “Relative amount of JEV RNA”. Just for reference, we have presented an alternative version of Fig. 5A calculated using the 2^-∆∆Ct method in a “supplemental file for reviewers 1”. The same conclusion was obtained with this method. 

Comment #15: Fig 6- Make all of the y-axes the same to better illustrate the shift from intracellular to extracellular JEV RNA at 2 and 3 dpi. Combine all 9 data points as they are all from 3 repeats of the same experiment.

Response: In accordance with the reviewer’s comment, we have made all of the y-axes the same, and all the data points have been combined in the revised Fig. 6. We have added the following sentences in the legend (p.22, L426�428): 

“Data represent the mean ± SD of three independent experiments. Each symbol represents a mean from one experiment”.

Comment #16: Fig 8A- Add statistics.

Response: We have added the results of statistical analysis to the revised Fig 8A and 8B. 

Comment #17: Fig 9- Please clarify how plaques are occurring at 4 dpi in Vero cells when no cell death is observed at this time-point (Figures 8C and 8D).

Response: A possible explanation for this discrepancy in cell death may have been due to differences in culture conditions: The serum concentration was 10% for Fig 8 and 2% for Fig 9. In addition, the medium for Fig 9 contained methylcellulose. These differences may have affected Vero cell death. In the case of JEV, the lower the serum concentration is, more cell death occurs (J Virol. 2005 Jul; 79 (13): 8388–8399). We have added this explanation and cited the paper (reference 27) in p. 24�25, L484�489. 

RESPONSES TO REVIEWER #2

 We are grateful to reviewer #2 for the critical comments and useful suggestions that have helped us to improve our paper. As indicated in the responses that follow, we have taken all these comments and suggestions into consideration in the revised version of our paper.

(Comments by reviewer #2 are shown in italics.)

Reviewer #2: In the present study of “Comparative characterization of flavivirus production in two cell lines: Human hepatoma-derived Huh7.5.1-8 and African green monkey kidney-derived Vero”, Saito and colleagues compared the JEV and YFV infection in they newly developed Huh7.5.1-8 and Vero cell lines. They showed that Huh7.5.1-8 cell produced more virus particles and more susceptible to virus-induced cell death than Vero cells upon JEV and YFV infection. They further showed that this is resulted from rapid viral replication kinetics and efficient virus release but not because of higher infection efficacy. They also showed that Huh7.5.1-8 cells developed plaques more rapidly than Vero cells after JEV infection and higher numbers of plaques after YFV infection than that of Vero cells. They finally showed the Huh7.5.1-8 cell containing a functional loss of the antiviral RIG-I gene. The authors showed that Huh7.5.1-8 cells are an useful substitute cell lines for studying viruses of Flaviviridae family.

Major Comment #1: About testing the extracellular JEV RNA in Huh7.5.1-8 and Vero cell lines in figure 5B, the authors expressed the extracellular JEV RNA as the total copy number in the culture supernatant divided by that of GAPDH RNA in the whole cells from the same well. However, the authors showed that JEV infected Huh7.5.1-8 cells showed significant less cell viability in figure 3 compared with that in Vero cells. This makes the normalized GAPDH in Huh7.5.1-8 cells is much less than that in Vero cells which will increase the values in Huh7.5.1-8 cells and also affect the interpretation showed in figure 6. I think it ‘s better to compare the total RNA copy number of the extracellular RNA between two cell lines because the same cells were initially seeded and the same volume of medium was used.

Response： In accordance with the reviewer’s comment, we have changed the measure in Fig 5B from “the extracellular JEV RNA as the total copy number in the culture supernatant divided by that of GAPDH RNA in the whole cells from the same well” to “JEV RNA as the total copy number in the supernatant”.

Because the other reviewer suggested to combine the results from multiple experiments, data from four experiments (in copies/well) have been normalized to that of Vero cells at 0 h pi in the same experiment, and then combined into one figure. Along with the integration of data, we have changed the following sentence from p. 20, L378�379 in the original version:

“…, whereas the rise for Vero cells was at 12 h pi (Fig. 5A [a])”

to

 “…, whereas the rise for Vero cells was not remarkable during 0–12 h pi (Fig. 5A [a]) ”(p. 20, L385�386). 

We have also changed the following sentence from p. 20, L380–382 in the original version:

 “Throughout the infection, Huh7.5.1-8 cells accumulated more viral RNA intracellularly (Fig 5A [b]) and extracellularly (Fig 5B [b]) than Vero cells” 

to

“Although the level of intracellular viral RNA was not different between the two cell lines after 12 h pi (Fig 5A [b]), the level of extracellular viral RNA was higher for Huh7.5.1-8 cells than for Vero cells during 12�48 h pi (Fig 5B [b])”(p. 20, L387�390). 

Major Comment #2: The codon 55 of the RIG-I gene Huh-7.5.1 is heterozygous or not? How do we know the heterozygous of codon 55 in the RIG-I gene is stable during passages? The authors should discuss about this.

Response: In the revised Fig 10, we have added the results of direct sequencing of codon 55 of the RIG-I gene from Huh7.5.1 cells and found that the codon of Huh7.5.1 cells was heterozygous as was that of Huh7.5.1-8 cells. This result indicates that the heterozygosity of codon 55 of RIG-I was stable at least during the time span for isolation of the Huh7.5.1-8 clone from Huh7.5.1 cells. We have discussed this in the revised manuscript (p. 30, L619�622). 

Minor Comment #1: For comparing the JEV production in Huh7.5.1-8 and Vero cell lines in figure 2 E and F, it seems the ratio in a bipolar distribution in 2 dpi in both high confluence and low confluence and day 1 in low confluence. Why the variation is so big?

Response: In the revised manuscript, we have changed the method of data combination from “ratio of JEV titer of Huh7.5.1-8 cells to that of Vero cells at each time point” (the original Figs 2E and 2F) to “the relative titer to the value of Vero cells at 1 d pi” (the revised Fig 2A and 2B) to show time-dependent changes of JEV titer of both cells. The large variations that the reviewer mentioned were still found after this revision, although the variation at 1 d pi under low confluency appeared less remarkable when plotted on a log scale. In addition, large variations between experiments were seen also with Fig 8 (YFV titer). Because these variations could not be linked to specific experimental conditions, we do not have convincing answers to the reviewer’s comment. Although the reason for the variations remains unknown, the data (Figs 2 and 8) definitely showed that JEV and YFV titers of Huh7.5.1-8 cells were significantly higher than those of Vero cells at early infection times.

Minor Comment #2: For figure 2 to indicate the high confluence and low confluence, its better to labeled as high or low confluence MOI 0.01/0.1. The way the authors labeled is easy to be considered as high MOI and low MOI.

Response: The reviewer’s comment is correct. We have deleted confusing labels such as “high MOI and low MOI”. In addition, we have separated the data for MOI 0.01 and MOI 0.1 into different figures (Figs 2 and S5 Fig) and have labeled them as high or low confluency.

Minor Comment #3: For comparing the infectivity of JEV in Huh7.5.1-8 and Vero cell lines in figure 4B, same question as above, the infectivity ratio in 2- and 3- dpi in a bipolar distribution which shows big variation.

Response: At present, we do not have convincing answers to the reviewer’s comment. However, we speculate that at least the bipolar distribution of infectivity ratio at 3 d pi was partly caused by non-infectious viral RNA leaking from dead Huh7.5.1-8 cells. If such a leak were poorly controlled, this could lead to variation in the infectivity ratio (the original Fig 4) and the particle-to-PFU ratio (the revised Fig 4). We have incorporated this discussion as follows (p. 19, L356�358):

“The reason for large variations found at 2 and 3 d pi remains unknown, but that at 3 d pi may have been partly due to variable amounts of non-infectious viral RNA leaking from dead Huh7.5.1-8 cells (Fig 3)”.

 

OTHER CHANGES

About the text:

1. The affiliation of Ryosuke Suzuki has been changed from “Department of Virology II, National Institute of Infectious Diseases, Shinjuku-ku, Tokyo, Japan” to “Department of Virology II, National Institute of Infectious Diseases, Musashi-murayama-shi, Tokyo, Japan” (p. 1, L 13�14). The affiliation of Naoki Osada has been changed from “Graduate School of Information Science and Technology, Hokkaido University, Sapporo, Hokkaido, Japan” to “Faculty of Information Science and Technology, Hokkaido University, Sapporo, Hokkaido, Japan” (p. 1, L 15�16).

2. To avoid misunderstanding, we have changed the following sentence (p. 2, L45�48) in the original Abstract:

“Huh7.5.1-8 cells were found to express not only a full-length RIG-I mRNA with a known dominant-negative missense mutation but also variants lacking exon 5/6�12 without the mutation, indicating functional loss of the antiviral RNA helicase RIG-I in the cells.”

to

 “Sequence analysis of cDNA encoding an antiviral RNA helicase, RIG-I, showed that Huh7.5.1-8 cells expressed not only a full-length RIG-I mRNA with a known dominant-negative missense mutation but also variants without the mutation. However, the latter mRNAs lacked exon 5/6�12, indicating functional loss of RIG-I in the cells.” (p. 2�3, L46�50).

3. Although the range of sampling points expressed in d pi was described in each legend in the original manuscript, they are summarized in the Material and methods in the revised manuscript (p. 7, L148�151). 

4. We have changed the following sentences from the original manuscript (p. 8, L168�171):

“After fixation and staining, plates were scanned with a document scanner, and image data were obtained. To improve plaque visibility, an invert filter was applied to the image data using Photoshop Elements (version 14; Adobe Systems, San Jose, CA, USA)”

to

“After fixation and staining, plates were scanned with a document scanner, and image data were obtained (S11 Fig). To improve plaque visibility, an invert filter was applied to the image data, and highlights were adjusted using Photoshop Elements (version 14; Adobe Systems, San Jose, CA, USA)” (p. 8, L174�177).

5. In the original manuscript, we mentioned unpublished results on p. 14, L234�237, p. 25, L497�500, and p. 26, L513�514. A paper including the results has recently been submitted to bioRxiv (Kawamoto M, Yamaji T, Saito K, Satomura K, Endo T, Fukasawa M, et al. Identification of characteristic genomic markers in human hepatoma Huh7 and Huh7.5.1-8 cell lines. bioRxiv. 2020:2020.02.17.953281. doi: 10.1101/2020.02.17.953281.). We have cited this paper in the revised manuscript (Reference No. 25). 

6. We have added the following sentence to p. 14, L245�246: 

“All the experiments were done with three biological replicates and repeated as described in the figure legends”.

7. We have deleted the following sentence from the original manuscript (p. 14, L245�246):

 ”Values of p < 0.05 were considered statistically significant”.

8. We have changed “among” to “between” (p.15, L255 and p. 16, L280).

9. We have deleted the following sentence from the original manuscript (p. 27, L553�554):

“�, which was remarkable particularly in high-cell-density infection”. 

10. We have changed the following sentence from the original manuscript (p. 29, L584�587):

“Vero cells may be more suitable to viruses requiring long-term cultivation to grow plaques, because Huh7.5.1-8 cells were less tolerant to our plaque assay conditions than Vero cells.”

to

“Vero cells may be more suitable to viruses requiring long-term cultivation to grow plaques, because they were more tolerant to our plaque assay conditions than Huh7.5.1-8 cells” (p.30, L608�611).

11. We have changed the text and the legends and to fit the revised figures.

About the figures:

12. One point (closed circle at 3 d pi) in our original Fig 2F was the wrong value (0.263): The correct value is 2.08. In addition, the p-value at 1 d pi (0.033) was wrong and has been corrected to 0.032. We sincerely apologize for these corrections. We have attached the corrected original Fig 2 as “Supplemental file for reviewers 2”, because this figure is not included in the revised manuscript. We hope the editor and reviewers could refer to the corrected figure. 

13. In Fig 2 of the original manuscript, we showed the results for JEV production and cell viability of Huh7.5.1-8 and Vero cells at MOI 0.01. In the revised manuscript, we have moved the results to a supplemental figure (S5 Fig), because we did not use the infection condition at MOI 0.01 in any other experiments. We have also changed the following text (p. 7, L140) :

“a multiplicity of infection (MOI) of 0.01 or 0.1 ” 

to

“a multiplicity of infection (MOI) of 0.1” (p. 7, L143).

14. In the original S2 Fig on JEV production of different Vero sublines, data drawn by line graphs were overlapping and difficult to make out. So, we have changed the data to bar graphs and have added the results of statistical analysis in the revised S4 Fig.

15. We have added unprocessed image data for virus plaques (S11 Fig). 

Correspondence table of original and revised figures:

Original Revised Changes

Fig 1A, 1B Fig 1A, 1B Results of multiple experiments have been combined with statistics.

Fig 1C Fig 1C, S3 Fig One experiment is shown in Fig 1C, and two other independent experiments are shown in S3 Fig. 

Fig 2 Fig 2, S5 Fig Results of multiple experiments have been combined with statistics. Results for MOI 0.01 have been moved to S5 Fig.

Fig 3 Fig 3, S5 Fig Results of multiple experiments have been combined with statistics. Results for MOI 0.01 have been moved to S5 Fig.

Fig 4 Fig 4, S6 Fig The measure has been changed from PFU per copy number to particle-to-PFU ratio. A representative result has been moved to S6 Fig. 

Fig 5 Fig 5 Results of multiple experiments have been combined with statistics. The measure has been changed from JEV/GAPDH to the relative mount of JEV RNA.

Fig 6 Fig 6 Results of multiple experiments have been combined with statistics.

Fig 7 Fig 7, S7 Fig One experiment is shown in Fig 7, and two other independent experiments are shown in S7 Fig.

Fig 8A, 8B Fig 8A-D, S8 Fig Representative experiments are shown in Fig 8A and 8B, and two other independent experiments are shown in S8 Fig. The combined results with statistics are also shown in Fig 8C and 8D. 

Fig 8C, 8D Fig 8E, 8F Results of multiple experiments have been combined with statistics.

Fig 9 Fig 9, S9 Fig One experiment is shown in Fig 9, and two other independent experiments are shown in S9 Fig.

Fig 10 Fig 10 Data for Huh7.5.1 cells have been added.

S1 Fig S1 Fig The results of statistical analysis have been added to S1A Fig.

S2 Fig S4 Fig The figure number has been changed. The original line graphs have been changed to bar graphs with statistics.

S3 Fig S10 Fig The figure number has been changed.

None S2 Fig Data for cell growth rates are shown. 

None S11 Fig Unprocessed image data (Figs 1C, 7, 9, S1B, S3, S7, and S9) are shown.

---

## [Editor Report · Decision Letter 1]

13 Apr 2020

Comparative characterization of flavivirus production in two cell lines: Human hepatoma-derived Huh7.5.1-8 and African green monkey kidney-derived Vero

PONE-D-19-32355R1

Dear Dr. Saito,

We are pleased to inform you that your manuscript has been judged scientifically suitable for publication and will be formally accepted for publication once it complies with all outstanding technical requirements.

With kind regards,

Negin P. Martin, Ph.D.

Academic Editor

PLOS ONE
---

## [Editor Report · Acceptance letter]

16 Apr 2020

PONE-D-19-32355R1 

Comparative characterization of flavivirus production in two cell lines: Human hepatoma-derived Huh7.5.1-8 and African green monkey kidney-derived Vero 

Dear Dr. Saito:

I am pleased to inform you that your manuscript has been deemed suitable for publication in PLOS ONE. Congratulations! Your manuscript is now with our production department. 

With kind regards,

on behalf of

Dr. Negin P. Martin 

Academic Editor

PLOS ONE